**Data Availability Statement:** All relevant data are within the paper and its Supporting information files.

# The effects of simple graphical and mental visualization of lung sounds in teaching lung auscultation during clinical clerkship: A preliminary study

Ayaka Kuriyama[1⊙], Hajime Kasai[1,2,3⊙]*, Kiyoshi Shikino[4], Yuki Shiko[5], Chiaki Kawame[1], Kenichiro Takeda[1,2], Hiroshi Tajima[1,3], Nami Hayama[1], Takuji Suzuki[1], Shoichi Ito[2,3]

1 Department of Respirology, Chiba University Graduate School of Medicine, Chiba, Japan, 2 Health Professional Development Center, Chiba University Hospital, Chiba, Japan, 3 Department of Medical Education, Chiba University Graduate School of Medicine, Chiba, Japan, 4 Department of General Medicine, Graduate School of Medicine, Chiba University, Chiba, Japan, 5 Biostatistics Section, Clinical Research Center, Chiba University Hospital, Chiba, Japan

⊙ These authors contributed equally to this work.
* daikasai6075@yahoo.co.jp

## Abstract

### Introduction

The study aimed to evaluate visualization-based training's effects on lung auscultation during clinical clerkship (CC) in the Department of Respiratory Medicine on student skills and confidence.

### Methods

The study period was December 2020–November 2021. Overall, 65 students attended a lecture on lung auscultation featuring a simulator (Mr. Lung™). Among them, 35 (visualization group) received additional training wherein they were asked to mentally visualize lung sounds using a graphical visualized lung sounds diagram as an example. All students answered questions on their self-efficacy regarding lung auscultation before and after four weeks of CC. They also took a lung auscultation test with the simulator at the beginning of CC (pre-test) and on the last day of the third week (post-test) (maximum score: 25). We compared the answers in the questionnaire and the test scores between the visualization group and students who only attended the lecture (control group, n = 30). The Wilcoxon signed-rank test and analysis of covariance were used to compare the answers to the questionnaire about confidence in lung auscultation and the scores of the lung auscultation tests before and after the training.

### Results

Confidence in auscultation of lung sounds significantly increased in both groups (five-point Likert scale, visualization group: pre-questionnaire median 1 [Interquartile range 1] to post-

**Funding:** The author(s) received no specific funding for this work.

**Competing interests:** The authors have declared that no competing interests exist.

questionnaire 3 [1], $p<0.001$; control group: 2 [1] to 3 [1], $p<0.001$) and was significantly higher in the visualization than in the control group. Test scores increased in both groups (visualization group: pre-test 11 [2] to post-test 15 [4], $p<0.001$; control group: 11 [5] to 14 [4], $p<0.001$). However, there were no differences between both groups' pre and post-tests scores ($p = 0.623$).

## Conclusion

Visualizing lung sounds may increase medical students' confidence in their lung auscultation skills; this may reduce their resistance to lung auscultation and encourage the repeated auscultation necessary to further improve their long-term auscultation abilities.

## Introduction

Lung auscultation is a fundamental part of physical examination; it is routinely performed by doctors and other medical professionals alike. Lung auscultation is a useful non-invasive method for clinical reasoning and identifying a respiratory condition using only a stethoscope [1]. Regarding clinical respiratory examination skills, auscultation of lung sounds is often learned during medical school [2]. However, it is not easy to accurately listen to lung sounds and assess respiratory conditions. In studies comparing the auscultation skills of medical students and physicians, there was no difference in skill, except for pulmonologists [3, 4]. This finding suggests that physicians may not have better lung auscultation skills than medical students. Therefore, medical students need more effective education programs on lung auscultation to improve their skills continuously.

Recent advances in digital stethoscopes have made it possible to record lung sounds and share them among multiple medical professionals, as well as to analyze lung sounds automatically, perform auscultation wirelessly, and provide telemedicine [5–7]. Moreover, in recent years, video-sharing sites on the Internet have made it possible to listen to lung sounds. However, the number of appropriate teaching materials is limited [8]. Even for pulmonary auscultation, web-based materials and computer applications on pulmonary auscultation are available [9–12]. However, effective education on how to perform lung auscultation has not been developed. The skills of lung auscultation have been taught using simulators, in addition to textbooks and audio sources [13–17]. These simulators can improve learner confidence and knowledge of lung sound auscultation [13, 14, 16]. Bernardi et al. examined the effects of simulator-based education on lung sounds and heart sounds (Cardiology patient simulator "K Plus" training system, Model #11257–159, Kyoto Kagaku Co. Ltd., Kyoto, Japan) and observed an improvement in the ability of auscultation of heart sounds but no change in that of lung sounds [15]. The reason for this result could be that the participants were asked to choose the relevant lung sounds from a list of options, while the heart sounds were answered graphically. In textbooks, lung sounds are often presented in simplified diagrams [1, 18]. The auscultation ability of some lung sounds has been improved by illustrating the mechanism that produces abnormal lung sounds [9]. Therefore, the addition of visual information regarding lung sounds may improve lung auscultation learning.

Lung sounds have been visualized by spectrograms, which have been published in textbooks, with the vertical axis representing sound pitch and the horizontal axis representing time [19]. In the spectrogram, intermittent rales are represented by short vertical lines and continuous rales are represented by horizontal lines. It is useful to learn the characteristics of

lung sounds [20, 21]; the use of spectrograms can aid medical students in diagnosing lung sounds and identifying abnormal lung sounds [21, 22]. However, spectrograms may not be available in all institutions, and it is difficult to visualize them with a normal personal stethoscope. Therefore, we hypothesized that a simplified diagram of lung sounds would improve lung auscultation ability by allowing the learner to visualize lung sounds without special devices.

We created a simple graphical and mental visualization of lung sounds using a combination of lines and circles in lung auscultation. Therefore, this study aimed to evaluate the effect of educational methods of lung sound auscultation using visualization for medical students in clinical clerkship (CC).

## Materials and methods

### Setting

**Pre-clerkship course and CC.** In Chiba University's school of medicine, each grade comprises around 120 students. Medical schools in Japan offer a six-year curriculum, with the final two years generally spent in CCs [23]. In our university, the CC starts in December of the fourth year and ends in October of the sixth year. In the third year, before the CC, students attended one 60-minute lecture on lung auscultation using a simulator (Mr. Lung®, Kyoto Kagaku Co. Ltd., Kyoto, Japan) as preparation for CC. Mr. Lung® is a mannequin-type lung sounds auscultation simulator widely used in medical schools in Japan (S1A Fig) [14, 16]. The purpose of the lecture was to help learners practice lung auscultation and understand the mechanisms of lung sounds. Four simulators were available; about six students performed auscultation using one simulator to practice listening to various lung sounds with a stethoscope. Subsequently, each student performed self-learning using a textbook and/or the simulator in preparation for the objective structured clinical examination (OSCE), which is administered in the fall of the fourth year before CC.

In this study, the number of participants was based on realistic possibilities without calculating the sample size. Students are rotated from one department to the other every four weeks in the CC of our university hospital. The sample for this study was recruited based on 120 students who underwent the CC in 2020–2021. Students in the following academic years had different backgrounds, as they learned lung auscultation only through e-learning before their CC due to the COVID-19 pandemic. In addition, the CC in the respiratory unit at our hospital was elective between Respiratory Medicine and Thoracic Surgery. Of the 120 students in the same year of the curriculum, 89 chose the Department of Respiratory Medicine. In our department's CC, groups of seven to eight medical students underwent a four-week training program as members of a medical team of doctors and residents. All medical students were assigned two to four patients during this four-week period and performed daily physical examinations, including auscultation of the lungs.

### Study design and samples

This single-center study included a cohort of 89 medical students who underwent CC in Respiratory Medicine at Chiba University Hospital between December 2020 and November 2021. Informed consent was procured from the participants during CC orientation to use their test results regarding lung auscultation and their questionnaire responses. Students who did not answer all parts of the questionnaire or take the lung auscultation test were excluded from the study. The 18 students who practiced in February and September 2021, during the COVID-19 pandemic in Japan, were also excluded, since the lectures and tests could not be held due to hospital access restrictions.

This study was approved by the ethics committee of Chiba University (approval number 3425). The study database was anonymized.

## Procedure of education on lung sound auscultation with visualization

The lecture on lung auscultation lasted for an hour and was presented to all students on the first day of CC in Respiratory Medicine. It consisted of an explanation of the mechanism of hearing lung sounds and diseases, as well as auscultation of various lung sounds by students with the simulator (Mr. Lung®, Kyoto Kagaku Co. Ltd., Kyoto, Japan).

The medical students were divided into two groups: a visualization group in which the participants attended the lecture and visualized the lung sounds, and a control group in which students only attended the lecture (Fig 1). The two groups were formulated on a rotational basis (Visualization group: January, April, June, October, November; Control group: December, March, May, July).

The students in the visualization group received approximately 30 minutes of additional training using a diagram of graphically visualized lung sounds during the lecture on lung auscultation in CC, as shown in Fig 2 and S2 Fig. In the diagram, the vertical axis represents the pitch of lung sounds, and the horizontal axis represents the duration of lung sounds as well as a spectrogram of lung sounds. The respiratory sounds were indicated by a right ascending diagonal line for inspiration and a right descending diagonal line for expiration [1]. Moreover, the thickness of the line represents the loudness of the sound. For the intermittent rales, each crackle is represented by a circle, and the height of the circle's position indicates the pitch of the sound, while the size of the circle indicates the loudness of the sound. The circular shape was chosen as coarse crackles and fine crackles because the former was caused by the bursting of blisters in the bronchi and the latter by the opening of obstructed peripheral bronchi. Thus,

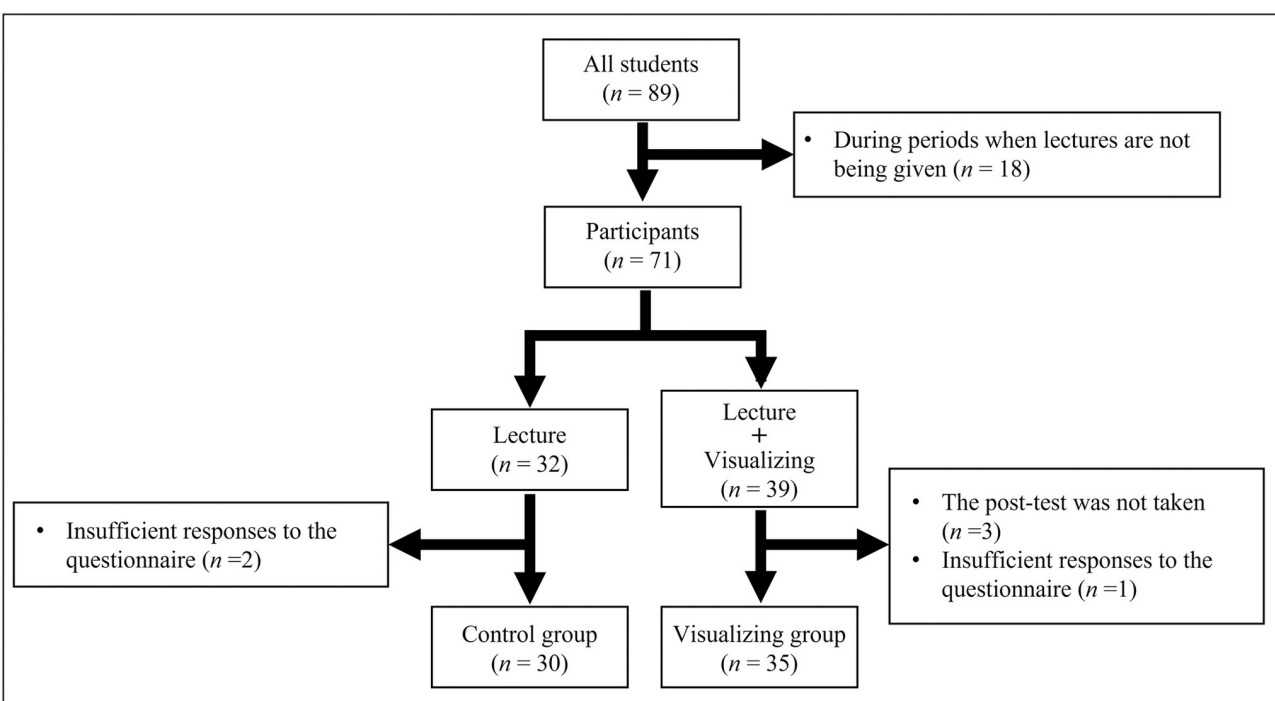

**Fig 1. Flow chart of the participants' selections and groups.**

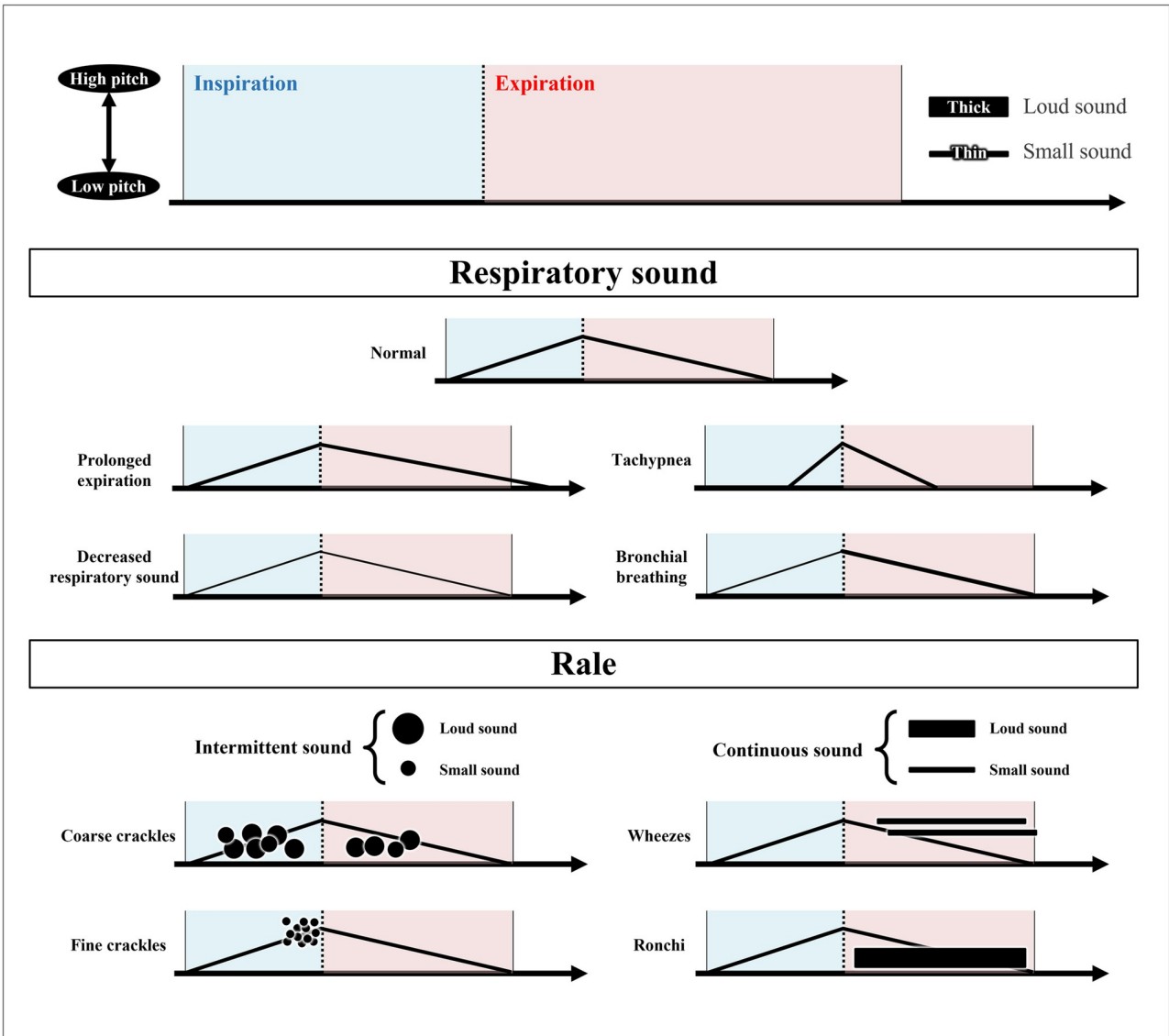

**Fig 2. The graphical visualization of lung sounds using a combination of lines and circles.** A diagram showing inspiration and expiration was used to visualize the lung sounds. In the diagram, the vertical axis represents the pitch of lung sounds and the horizontal axis represents the duration of lung sounds. For respiratory sounds, the thickness of the line represents the loudness of the sound. For the intermittent rales, each crackle is represented by a circle, the height of position indicates the sound's pitch, and the circle's size indicates the loudness of the sound. Thus, coarse crackles are presented as circles drawn at a lower position on inspiration and/or expiration. Fine crackles are presented as circles that are drawn at the end of the inspiration and are concentrated at a high position. For continuous rales, a horizontal bar was used. The height of the position of the horizontal bar indicates the pitch of the sound, and the thickness of the bar indicates the loudness of the sound. Wheezes are mainly represented by a horizontal bar at the high end of the expiration and rhonchi are also mainly represented by a horizontal bar at the low end of the expiration.

coarse crackles were presented as circles that are drawn at a lower position on inspiration and/ or expiration. Fine crackles are presented as circles that are drawn at the end of the inspiration and are concentrated at a high position. The height of the horizontal bar position indicates the pitch of the sound, and the thickness of the bar indicates the loudness of the sound as well as the continuous sound shown in the spectrogram. Wheezes are mainly represented by a horizontal bar at the high end of the expiration, and rhonchi are also mainly represented by a

horizontal bar at the low end of the expiration. The diagram was created by consensus between Respiratory Medicine specialists (HK) and medical education specialists (KS).

During the lecture, the students drew the various lung sounds they listened to on a blank diagram. Then, the students were instructed to mentally visualize lung sounds while performing lung auscultation for patients.

### Evaluation of the effects of education on lung sound auscultation with visualization

**Questionnaire.** To collect quantitative data, participants completed a questionnaire before and after the CC in Respiratory Medicine (S1 Table, Fig 3). Students were asked to rate their own lung sound auscultation ability in questions A1 and B1, whether they had any experience in listening for various lung sounds in questions A2 and B2, and their confidence in auscultating lung sounds in questions A3 and B3. In addition, after the CC, they were asked

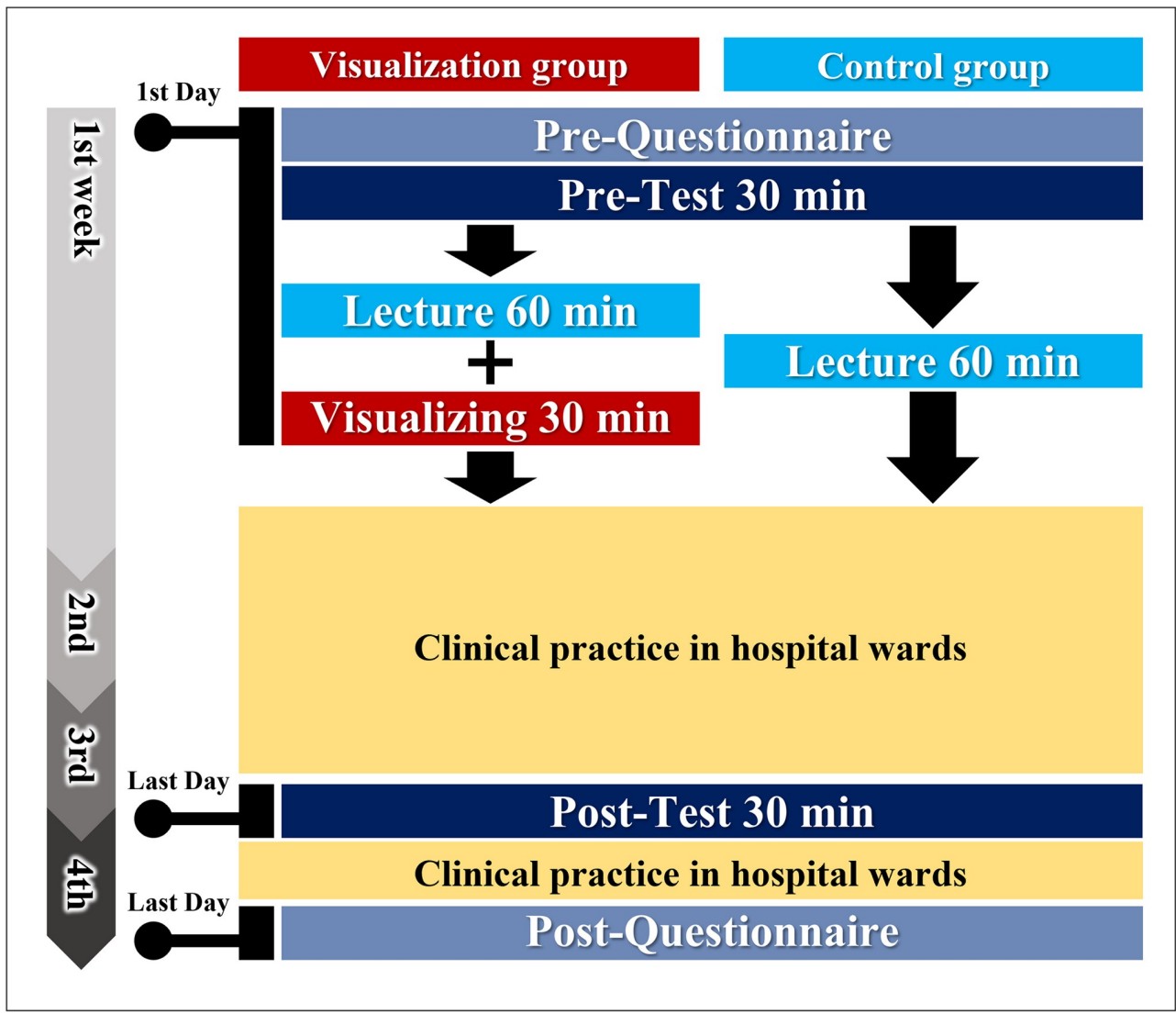

**Fig 3. Protocol for lung auscultation education and questionnaire and test regarding lung auscultation.**

**Table 1. Abnormal lung sounds and where they were listened, as used in the lung auscultation examination.**

| Case of abnormal lung sounds | Location |
| --- | --- |
| Decreased breath sound | Left lower lung field |
| Coarse crackles | Right upper middle lung field |
| Coarse crackles | Bilateral upper lung fields |
| Fine crackles | Bilateral lower pulmonary region |
| Fine crackles | Bilateral whole lung fields |
| Wheezes, Prolonged expiration | Bilateral upper lung fields |
| Ronchi, Prolonged expiration | Bilateral upper lung fields |
| Wheezes, Ronchi, Prolonged expiration | Bilateral upper lung fields |
| Squawk, Coarse crackles | Right upper and middle lung fields |
| Pleural friction rub | Right middle and lower lung fields |

The pre- and post-tests consisted of the same ten sounds, while each case was given in a different order.

about their satisfaction level with the educational program on lung auscultation in question B4. Indications 1 and 5 were presented in the actual questionnaire to the students. In addition, although not indicated on the questionnaire form, it was explained orally that 3 meant fair. For questions A2, A3, B2, and B3, the students responded to nine types of lung sounds based on the classification of Bohadana et al. [24].

**Performance of lung auscultation.** Students' lung auscultation performance was evaluated by performing an auscultation test of 10 cases using the Mr. Lung™ simulator (S1B Fig). The simulator was used to evaluate the relevant auscultation skills with an awareness of the location and timing of breathing rather than simply playing sounds. The pre-test was performed at the beginning of CC in Respiratory Medicine before the lecture. The post-test was performed on the last day of the third week of the CC. The pre- and post-tests consisted of the same 10 sounds, while each case was given in a different order (Table 1). In each case, one point was given for each correct answer of lung sound abnormality and location. In four cases, in which more than one abnormal lung sound could be heard, one point was given for each sound. Therefore, the maximum score was 25 points. Both the pre-test and post-test taken by the control group were multiple-choice forms (S1C Fig). The pre-test taken by the visualization group was a multiple-choice form, while the post-test was a multiple-choice form that included a space to draw a figure where there was an abnormality (S1D Fig). Students were not limited in time and number for lung auscultation while answering each question in pre- and post-tests. Moreover, students used the diagrams on the answer sheet as an aid in answering the questions. The diagrams students drew were not included in the evaluation.

We then compared the answers in the questionnaire and the test scores between the visualization group and control group.

**Statistical analysis.** All results are expressed in terms of the median and interquartile range (IQR) unless otherwise indicated. Normality was evaluated by performing the Shapiro–Wilk test. In the data collected in this study, the pre- and post-tests for the control group were normally distributed, while the other parameters were non-normally distributed (S2 Table). The Wilcoxon signed-rank test was used to compare the questionnaire answers about participants' confidence in lung auscultation and the test scores of lung auscultation before and after the training. Confidence in and scores on lung auscultation in the two groups before and after our lecture were compared using analysis of covariance (ANCOVA), by adjusting the values of the pre-questionnaire and pre-test results for each group. Change of confidence in and scores on lung auscultation in the two groups before and after our lecture were compared using

multivariate linear regression. In the multivariate analysis, the values of the pre-questionnaire and pre-test results were adjusted. A $p$-value < 0.05 was considered statistically significant. All statistical analyses were performed using JMP 15.0 software (Cary, North Carolina, USA) and SAS software version 9.4 (SAS Institute, Cary, USA).

## Results

In total, 71 students completed the lecture for auscultation of lung sounds. Of these students, three did not take the post-test and were therefore excluded from the study. Another three students were also excluded because their questionnaires had insufficient data. Finally, 65 students were included in the study. The visualization group included 35 students (age median 23 [IQR 2]; male/female 27/8), and the control group included 30 students (22 [IQR 1] years; male/female 24/6) (Fig 1). There were no significant differences in the age and sex ratio ($p = 0.123$ and $p = 0.780$, respectively). In addition, there was no significant difference in the results of CC in Respiratory Medicine, which did not include the results of the lung auscultation test between the two groups (maximum score: 100, 76.4 [5.8] vs. 76.5 [5.4], $p = 0.802$).

### Questionnaire

Based on an analysis of their responses, students' satisfaction level with our education was high in both groups, but there was no significant difference between groups (visualization group: median 5 [IQR 1] vs. control group: 4 [1], $p = 0.150$). There was no significant difference between the two groups in the experience of listening to each lung sound at the beginning and end of CC in Respiratory Medicine (A2 and B2: "Have you ever heard each lung sound from a patient?") (S2 Fig).

   Confidence in lung auscultation (A1 and B1: "How is your current lung sound auscultation ability?") significantly increased in both groups (five-point Likert scale; visualization group, pre-questionnaire: 1 [1] to post-questionnaire: 3 [1], $p < 0.001$; control group, 2 [1] to 3 [1], $p < 0.001$). Confidence in lung auscultation after CC was found to be significantly higher in the visualization group than in the control group ($p = 0.028$; adjusted mean difference between pre- and post-questionnaire, 1.7 [standard error 0.1] vs. 1.3 [0.1], $p = 0.020$). In contrast, there was no difference between the two groups in self-assessment of listening to each lung sound (A3 and B3: "How confident are you in each lung sound?") (S3 Fig).

### Results of tests of lung auscultation

The test score increased in both groups (visualization group, pre-test 11 [2] to post-test 15 [4], $p < 0.001$; control group, 11 [5] to 14 [4], $p < 0.001$) as shown in Table 2. There was no difference between the scores of the pre-and post-tests for both groups (pre-test, $p = 0.827$; post-test, $p = 0.290$). In addition, there was no change in scores between the two groups (adjusted mean difference between pre-and post-tests, 3.7 [standard error 0.5] vs. 3.1 [0.6], $p = 0.424$).

## Discussion

To the best of our knowledge, this is the first study to evaluate the effect of graphical and mental visualization of lung sounds in lung auscultation teaching during CC. There are two main findings. First, the visualization of lung sounds can improve medical students' confidence in their lung auscultation. Second, the visualization of lung sounds could not improve medical students' ability to auscultate lung sounds; further improvements are needed. Our method does not require any special tools and can be easily and quickly implemented into usual education.

**Table 2. Changes in the confidence level pertaining to lung auscultation and the score of lung auscultation before and after education (*n* = 65).**

| | Visualization group (n = 35) | Control group (n = 30) | p-value |
|---|---|---|---|
| Confidence in lung auscultation, mean (SD) | | | |
| Pre-questionnaire | 1 (1) | 2 (1) | 0.128 |
| Post-questionnaire | 3 (1) | 3 (1) | **0.028** |
| Mean difference | 2 (1) | 1 (1) | **0.005** |
| Adjusted mean difference, least mean square (SE) | 1.7 (0.1) | 1.3 (0.1) | **0.020**[*] |
| Score of lung auscultation, mean (SD) | | | |
| Pre-test | 11 (2) | 11 (5) | 0.827 |
| Post-test | 15 (4) | 14 (4) | 0.290 |
| Mean difference | 4 (4) | 3 (1) | 0.623 |
| Adjusted mean difference, least mean square (SE) | 3.7 (0.5) | 3.1 (0.6) | 0.424[*] |

[*] Analysis of covariance (ANCOVA) was performed, and pre-score was adjusted.

Standard error, SE.

Visualizing lung sounds in lung auscultation can boost medical students' confidence in lung auscultation. In the auscultation of heart sounds, the visualization of the timing of rhythm and murmur has been used [25]. Lung sounds have often been visualized in textbooks and reports [1, 18]. Sestini et al. reported that a multimedia presentation of acoustic and graphic characteristics of lung sounds could improve lung auscultation learning among medical students. In their study, the questionnaire completed by the students indicated that the association of the acoustic signals with their visual image was useful for learning and understanding lung sounds [20]. Higashiyama et al. reported that illustration could support understanding the generation mechanism with wheezes and coarse crackles and improve auscultation skills [9]. However, there have been few opportunities to visualize lung sounds in daily practice. In addition to lectures using spectrograms and sound sources as presented in the report, we evaluated how simple graphical and mental visualization of lung sounds affected the lung auscultation of medical students in CC. Although the primary objective was to distinguish abnormal lung sounds from normal lung auscultation in the patient, limited interest has been shown in determining the timing of breathing and/or the pitch of the abnormal sounds that can be heard. However, it is important to recognize the timing of breathing and the pitch of abnormal lung sounds for the evaluation of the respiratory status and for clinical reasoning. In other words, the visualization of lung sounds may have made lung auscultation more analytical, considering several elements of lung sounds in inspiration and expiration timing, pitch, and duration. Thus, a possible reason for the increase in confidence is that the visualization of lung sounds may have encouraged students to listen more carefully and consciously to the difference between inspiration and expiration and the mechanism behind each lung sound.

Moreover, there are four learning styles: visual, auditory, reading/writing, and kinesthetic [26]. The acquisition of auscultation skills can be referred to as an auditory style of learning. Visualization of lung sounds can make lung auscultation learning multimodal by combining visual and auditory styles, and this may help students be more aware of the timing and pitch of lung sounds. Additional clinical information in auscultation can help diagnose and distinguish different lung sounds [27, 28]. Furthermore, additional visual information, such as a spectrogram, can be useful in lung sound auscultation [21, 22]. Even if students could distinguish the lung sound correctly, this multifaceted approach using visualization may have led to students' perception that the pathophysiology could be better understood and analyzed. However, the detailed process for improving confidence auscultation in students was not completely clear in

this study, and further qualitative evaluation, such as focus group interviews and question-naires, will be necessary in future studies. Because of the lack of confidence, students may be hesitant to perform lung auscultation on patients. There is a possibility that students who are more confident in lung auscultation actively repeat lung auscultation more often. The repetitive practice of clinical skills can contribute to the retention of these skills [29]. In these students, auscultation skills may improve further in the long term.

Furthermore, it is unclear how students listen to lung sounds and how they evaluate them in real patients. By providing regular opportunities to capture lung sounds that can be heard using our method, it will be possible to clarify the gap between the findings of medical students and those of the supervising physician. As a result, lung auscultation can be effectively taught. In any case, our educational method of lung sound visualization does not require any additional tools, nor does it require much effort to implement. Although its effectiveness may be limited, it is a method that can be easily introduced into regular education and practice.

This study did not find any significant effect of visualization on lung auscultation. There was no significant difference between the post-test scores of both groups, and there was no change in scores between the two groups. Since post-test scores were only around half the maximum number of points, confidence in lung auscultation will not suffice. There are several possible reasons for the limited effect of our educational method. For the visualization group, we presented a visualization of lung sounds and asked students to draw a diagram of the lung sound that they heard during the lecture. Then, the students were also instructed to imagine lung sounds during their daily practice of lung auscultation. Only instruction to repeatedly visualize lung sounds may be insufficient. The study did not investigate how often students mentally imaged lung sounds during lung auscultation. Therefore, it is not clear how many students performed lung auscultation by visualizing lung sounds. Retention by instructing students to visualize lung sounds during their daily practice may have been more effective. Additional opportunities to draw lung sounds using the simulator or record lung sounds may have enhanced the effectiveness of our educational method. Furthermore, it may be necessary to instruct students to visualize lung sounds repeatedly during daily rounds with their supervisors and provide opportunities to draw visualizations at least once a week.

Moreover, for effective learning, reflective observation and abstract conceptualization are necessary, in addition to active experimentation and concrete experience. Repeating these four steps enhances learning [30]. Malmartel et al. examined the effect of combining a cardiopulmonary auscultation application (Medsounds™, Interactive Systems For Healthcare®) with clinical practice on medical students' auscultation of heart and lung sounds [12]. Similar to our study, they used a high-fidelity simulator to assess the auscultation of coarse crackles in pneumonia, showing that students' auscultation performance improved. In addition to the application, they suggested that appropriate supervision from the educator may have had an important impact.

Another reason for the lack of a significant effect in our study can be that the diagram for the visualization of lung sounds was not appropriate for learners to image mentally. The image of lung sounds can be different for each person; thus, it may be necessary to change the diagram to make it more intuitive [31]. For example, breath sounds may be better shown in a diagram combining triangles rather than lines, as shown in the following figure. Furthermore, it may be easier to intuitively visualize the coarse and fine crackles using an X instead of a circle (S5 Fig). In this study, students in the visualization group drew lung sound diagrams only as an aid during the post-test. In the future, students will be provided with multiple opportunities to draw lung sounds, and their lung sound diagrams will be evaluated.

## Study limitations

The present study has five main limitations. First, in terms of research design, this was a single-site study with a small number of participants in an uncontrolled environment and relied partly on students' self-assessments for data collection. Therefore, a larger number of participants based on the calculation of the required sample size and the use of a reliable questionnaire are required in the future. Second, the time spent on lung auscultation education differs between the two groups with and without time spent on training to visualize lung sounds. This difference in time may have affected students' confidence in auscultation as a confirmation bias. In addition, the time taken for the self-study of lung auscultation before and during CC can be different for each student. Third, it is unclear how often students continued to visualize lung sounds in their daily practice. Fourth, in the questionnaires for the students, only values 1 and 5 were defined in the 5-point Likert Scale of the question regarding confidence in lung auscultation. The other values were self-judged by the students. Thus, more rigorous definitions for all values are needed in future studies. Fifth, long-term effects have not been explored in our study.

## Conclusion

The visualization of lung sounds may improve medical students' confidence in lung auscultation. Although the visualization of lung sounds did not significantly affect the lung auscultatory ability to identify lung sounds, its advantages include the possibility of being undertaken without additional devices, such as special simulators, thus serving as a useful educational strategy. Further improvement in teaching methods may increase the learning effect of visualization. In the future, we are planning to conduct an auscultation education program using improved visualization of lung sounds and verifying the effectiveness of this method.

## Supporting information

**S1 Fig.** The simulator (A. Mr. Lung®, Kyoto Kagaku Co. Ltd., Kyoto, Japan) was used for lectures before CC and during CC in Respiratory medicine, the pre- and pos- test (B). Answer sheet for lung sound auscultation test. A multiple-choice form (C) was used as the pre-test for the control and visualization group and the post-test for the control group. A multiple-choice form that combined a space in which to draw a figure where there was an abnormality (D) was used as the post-test for the visualization group. CC, clinical clerkship.
(TIF)

**S2 Fig. Details of the lecture on lung auscultation conducted in the clinical clerkship and additional training on the visualization of lung sounds.**
(TIF)

**S3 Fig. The responses of the visualization group (*n* = 35) and the control group (*n* = 30) regarding the experience of listening to each lung sound.** C, control group; NS, not significant; V, visualization group.
(TIF)

**S4 Fig. The responses of the visualization group (*n* = 35) and the control group (*n* = 30) regarding self-assessment of listening to each lung sound.**
(TIF)

**S5 Fig. Proposed changes to lung sound visualization.**
(TIF)

**S1 Table. Questionnaire items before and after the training on lung auscultation.**
(DOCX)

**S2 Table.** *P*-value in the test of normality of each parameter (*n* = 65).
(DOCX)

**S1 File. De-identified data analysed in this study.**
(XLSX)

## Acknowledgments

We would like to thank Editage (www.editage.com) for English language editing.

## Author Contributions

**Conceptualization:** Ayaka Kuriyama, Hajime Kasai, Kiyoshi Shikino, Takuji Suzuki, Shoichi Ito.

**Data curation:** Ayaka Kuriyama, Hajime Kasai, Kiyoshi Shikino, Chiaki Kawame, Kenichiro Takeda, Hiroshi Tajima, Nami Hayama.

**Formal analysis:** Ayaka Kuriyama, Hajime Kasai, Kiyoshi Shikino, Yuki Shiko, Kenichiro Takeda.

**Investigation:** Ayaka Kuriyama, Hajime Kasai, Kiyoshi Shikino, Chiaki Kawame, Kenichiro Takeda, Nami Hayama, Shoichi Ito.

**Methodology:** Ayaka Kuriyama, Hajime Kasai, Kiyoshi Shikino, Yuki Shiko, Hiroshi Tajima.

**Project administration:** Ayaka Kuriyama.

**Supervision:** Kiyoshi Shikino, Yuki Shiko, Kenichiro Takeda, Hiroshi Tajima, Takuji Suzuki.

**Validation:** Hajime Kasai, Yuki Shiko, Hiroshi Tajima, Shoichi Ito.

**Visualization:** Hajime Kasai.

**Writing – original draft:** Ayaka Kuriyama, Hajime Kasai.

**Writing – review & editing:** Ayaka Kuriyama, Hajime Kasai, Kiyoshi Shikino, Yuki Shiko, Chiaki Kawame, Kenichiro Takeda, Hiroshi Tajima, Nami Hayama, Takuji Suzuki, Shoichi Ito.

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
