## [Decision Letter · Decision Letter 0]

10 Oct 2022

PONE-D-22-25852Does simple graphical and mental visualization of lung sounds improve the auscultation skills of clinical clerkship students?PLOS ONE

Dear Dr. Hajime Kasai,

Thank you for submitting your manuscript to PLOS ONE. After careful consideration, we feel that it has merit but does not fully meet PLOS ONE’s publication criteria as it currently stands. Therefore, we invite you to submit a revised version of the manuscript that addresses the points raised during the review process.

Your research is important, but the issues mentioned by the referees and especially emphasized need to be clarified. In this context, I think that the purpose of the research and the expectation should be explained by justifying it. Necessary adjustments should be made in your article so that the research methodology is defined more clearly and in detail and that the reader does not leave any questions in his mind. Results and the effect of "simple graphical and mental visualization of lung sounds" on "auscultation skills" should be expressed more clearly, and comments should be based on concrete data. The revisions you will make by reviewing all the suggestions will make your research more understandable and valuable.:Indicate which changes you require for acceptance versus which changes you recommendAddress any conflicts between the reviews so that it's clear which advice the authors should followProvide specific feedback from your evaluation of the manuscriptPlease ensure that your decision is justified on PLOS ONE’s publication criteria and not, for example, on novelty or perceived impact.

We look forward to receiving your revised manuscript.

Kind regards,

Ayse Hilal Bati, Associate Professor

Academic Editor

PLOS ONE

Journal Requirements:

Additional Editor Comments:

Dear Author/s,

Thanks for sharing your research with us. Your research is important, but the issues mentioned by the referees and especially emphasized need to be clarified. In this context, I think that the purpose of the research and the expectation should be explained by justifying it. Necessary adjustments should be made in your article so that the research methodology is defined more clearly and in detail and that the reader does not leave any questions in his mind. Results and the effect of "simple graphical and mental visualization of lung sounds" on "auscultation skills" should be expressed more clearly, and comments should be based on concrete data. The revisions you will make by reviewing all the suggestions will make your research more understandable and valuable.

Reviewers' comments:

Reviewer's Responses to Questions

**Comments to the Author**

1. Is the manuscript technically sound, and do the data support the conclusions?

Reviewer #1: Partly

Reviewer #2: Yes

Reviewer #3: No

2. Has the statistical analysis been performed appropriately and rigorously? 

Reviewer #1: I Don't Know

Reviewer #2: Yes

Reviewer #3: Yes

3. Have the authors made all data underlying the findings in their manuscript fully available?

Reviewer #1: No

Reviewer #2: Yes

Reviewer #3: Yes

4. Is the manuscript presented in an intelligible fashion and written in standard English?

Reviewer #1: Yes

Reviewer #2: Yes

Reviewer #3: Yes

5. Review Comments to the Author

Reviewer #1: Thank you for submitting this manuscript. The following are my questions/comments for the authors to address prior to resubmitting and another round of peer review:

I said I do not know if statistics were performed adequately because there is not enough detail in the statistical methods for me to determine if the tests were the correct ones. See info below about statistical analysis.

Data availability: you shared your analysis but not your data. please upload de-identified data to an online data repository to fulfill this requirement.

Lines 40-42: the p value is not significant, nor even close. Therefore you can't say that there was any difference between groups in score

Line 49: suggest adding simulation, model

Lines 56-59: I don't feel like these 2 sentences are relevant to your paper and would remove them from your intro. Possibly you could work them into your discussion? But because you used med students, it doesn't seem important how physicians and pulmonologists score relative to med students.

Lines 63-64: reference 6 refers to one simualtor. Do you have references for more simulators, for textbook training, and for other audio sources?

Line 88 - suggest changing grade to cohort. This is a better term to be internationally understood.

Lines 91-93: can you clarify how the lecture included a simulator? did the teacher use the simulator and the students watched? did the students use the simulator? if so, were they in groups or as individuals?

Lines 118-120: this study has an A vs A+B study design. Meaning lecture vs lecture + visualization training. This is a weak study design because it doesn't compare groups with equivalent time spent learning. We would expect students who spent more time in instruction (the visualization group) to be more confident and out-perform the lecture only group. A better study design is to compare 2 groups with equal time spent learning, and only the method of learning is different. Your study design with uneven educational time should be added to the study limitations.

Line 134 - "of the inspiratory"? I don't understand this wording.

Lines 135-136: this is repetitive

Lines 140- 152: this is repetitive and is too long for a figure caption. make the figure caption just the first sentence.

Lines 164-165: please define all steps of the likert scale, not just 1 and 5.

Lines 176-177: it would be good to remind the reader here how much time there was in between the lecture and the visualization training, and between that training and the performance test.

Lines 197-198: please describe how you tested your data for normality and what the result was.

Line 198: "compare the answers" - the answers to what?

Lines 212-213: this p value is not significant, nor even close, so you cannot say that there was any difference between groups in satisfaction level.

Lines 227-229: this p value is not significant, nor even close, so you cannot say that there was nay difference between groups.

Line numbering stopped at the discussion. Please correct this when resubmitting.

Discussion paragraph 1, sentence 4: you cannot say this because there was no difference between groups.

Discussion paragraph 2, sentence 1: you can't say that it may help their lung auscultation skills, because there was no difference between groups.

discussion paragraph 2: I would suggest having the sentence starting "Moreover," start a new paragraph.

discussion paragraph 2, sentence starting "since post-test scores were only...": so how do you suggest that this training can be improved so that students achieve better scores? how can the lecture be improved? how can the visualization exercise be improved?

discussion paragraph 3, sentence 1: you can't say this. the p value was not significant.

discussion paragraph 3, sentence 3: this needs to be highlighted as a limitation of the study. if the study was to evaluate non-visual training (lecture only) vs non-visual training (lecture) and visual training, then the non-visual portion should not have included visual training! this muddies the waters.

discussion paragraph 3, sentence starting "Furthermore...": this sentence does not make sense. please revise.

study limitations: you need to add the A vs A+B design, unequal time spent in teaching in the 2 groups, and that the lecture only group also got a discussion of visualization during the lecture

conclusions: sentence 1 - you cannot say that it may improve their auscultation skills - your p value was not significant.

general word on statistics - you seem to want to say that there are increases when the p value is insignificant. this is not acceptable. if you want to calculate effect size for the variables you measured, then you could say that the p value was not significant but the effect size was small/med/large and discuss implications - e.g. do you need a larger study to detect a different p value? Please consider adding effect sizes to your calculations. If you do not, all you can say is that there was no difference in any measure except confidence, because the p values are insignificant.

references 7-8: can you find more scientific references (e.g. journal articles) rather than citing sources such as youtube?

Reviewer #2: Kasai H et al present here the results of a study evaluating an interesting teaching method. Indeed, pulmonary auscultation is a fundamental clinical skill for any physician. The educational tool studied here also has the advantage of being simple and inexpensive.

The article is very well written, concise and presents the methodology (the pedagogical concept) and the results quite clearly.

Several comments and minor revisions should be made to the manuscript.

Introduction : another medical university evaluated the interest of simulation software in the learning of cardiopulmonary and published their results (PMID 32660857). The reference could be cited because evaluation of auscultation skill was done using a high-fidelity simulator after training students with a specific software.

Methods/results sections:

- The 3 questions asked to students (P7) might be presented in a table in the method section; results for each question might follow the same order in the result section (p10). Please change the order of questions or the order os answers and recall the question by putting it in brackets in the results section for each of the sentences detailing the answers. Reading and understanding of results will be facilitated.

- In the method section page 8 , table 1 appears in the text while it might appear with tables. Same comment for tittle and footprints for Fig2.

- In the method section "performance of lung auscultation" details that " multiple-choice form that included a space to draw a figure where there was an abormality". No results concerning the drawings proposed by the students during this post-test evaluation have been reported in the results section of the manuscript. Are they unexploited data or unexploitable data? Can you clarify this point, in particular how can we compare the graphic representations during the post-test evaluation with those made during the lecture?

- Method section : Did the students have to follow a particular examination plan (number and order of auscultatory foci/number of respiratory cycles per focus)?

I suggest presenting in a simple table (or in the text) some data (if available) concerning the students of each group: age, sex ratio, results of the faculty exams and grades/evaluations of the clinical clerkship.

Discussion section :

P 13, paragraph 1 The authors cannot conclude that "the visualization of lung sounds may improve student's ability" based on their results. It seems necessary to rephrase this conclusion.

P13, paragraph 2 "Although the primary objective was to identify...". This formulation is very confusing while identifiying abnomal lung sounds was not the primary objective of the study. I understand that, the main objective of the auscultation skill is to be able to distinguish abnormal sounds from normal auscultation.

p13, second last line : I suggest to start a new paragraph withe the sentence "Moroever...."

p14, the sentence "Since post-test scores were only..." could be moved to the beginning of the next paragrap.

p14, can the authors argue the sentence "However, the satisfaction level was also significantly higher in the visualization group, suggesting that visualization may have facilitated student's better understanding of lung sounds" ? To my knowledge, student's satisfaction with education/lecture/course is associated with a better understanding. References to support authors point of view are required.

p14, second paragraph : "However, it was clear how often the student continued to visualize lung sounds in their daily practice" should be moved to the limit paragraph (p15).

p14 : the last sentence "Firthermore" is not understable, please rephrase.

p15 : the second limit discussed here is for me one strength : in respiratory departments, students have access to a wide range of abnormal lung sounds. This is well shown par figure S3.

Reviewer #3: Dear editor,

I thank you for the invitation to review the manuscript “Does simple graphical and mental visualization of lung sounds improve the auscultation skills of clinical clerkship students?” The manuscript describes a study that explored the impact of graphical and mental visualization of lung sounds on the confidence and performance of lung auscultation in medical students. The study explored an interesting research problem that has not been explored in the past. The authors had a good idea of using graphical and mental visualization of lung sounds for learning of lung auscultation. In addition, the learning of the lung sounds was supported by a learning with a simulation model. However, the study has many weaknesses that impact negatively on its scientific quality.

My comments are here.

Regards

Comment.

The title must be written like a statement but not like a question. A question does not show the impact or the value of the study.

Abstract

Methods

The method section says that 79 students participated in the study. Hence, 35 students were part of the visualization group. But also, the method section says that 30 students were part of the control group. This is 65 students, but not 79 students. Authors must check this situation. Authors should include the name of the statistical test that they applied to perform the analysis of the data.

Introduction section

Comment

Page 3. Line 56. Please check the sentence. Authors say that “in a study”, but at the end of the sentence they included two references. Therefore, is not “in a study”, but “in studies”.

Comment

Page 3. Line 68. Bernardi et al…

In the abstract, the authors mention that they used a simulator (Mr. Lung), but in the Introduction section they did not mention something about that. Authors must include a brief description of the simulator and its possible impact on the learning of the students. In the Bernardi et al. study, do those authors also use Mr. Lung? Or other simulation model? In the present study, what is the relevance of using the simulation model? Authors should explain that in the Introduction section.

Comment

Page 4. Line 74. Is there any study exploring the learning of lung sounds by students with those spectrograms? If there is not, then authors can say that to give a better reason to perform its study. If there is some study, authors should mention it.

Comment

Page 4. Line 77. While it is useful…

Why the authors say that the spectrograms are not available in all institutions? Is there any evidence to support this statement? Maybe the statement should not be written as an absolute statement.

General comment. The research problem must be clearer, authors must support the research idea in the basis of the literature. The scientific background must be stronger.

An hypothesis must be included in the Introduction section.

Materials and methods section

Comment

Page 4. Line 92. “students attended a one-hour lecture of lung auscultation using a simulator”.

Is it one-hour per week? Or only one-hour for the CC?

Comment

Page 5. Line 102. “A total of 89 medical students underwent CC…”

The number of participants must be obtained from the calculation of the sample size “N”, thus authors must include the sample size and how they obtained the sample size.

Page 4, line 92. “Students attended a one-hour lecture on lung auscultation using a simulator”.

Is it one-hour per week? Or only one-hour for the CC? What simulation model was used by the students? Give information about the simulation model.

Comment

Page 4, line 94. Subsequently, each student performed self-learning using a textbook and/or the simulator in preparation for the objective structured clinical examination (OSCE), which is administered in the fall of the fourth year before CC.

Did the authors control the self-learning? Or the authors know the number of hours that each student spent on self-learning with a textbook? Or the authors know the number of hours that each student spent with the simulator in preparation for the OSCE? Because the distinct preparation of each student can influence the results.

Comment

Page 5. Line. “A total of 89 medical students underwent CC in Respiratory Medicine at Chiba University Hospital between December 2020 and November 2021”

The number of participants must be obtained from the calculation of the sample size (“N”). Thus, authors must include the sample size and how they obtained the sample size. The reason that the authors had 89 medical students in the CC is not a reason to support the participation of the 89 medical students. The calculation of sample size is mandatory for this type of study.

Comment

Page 5. Line 116. “Mr Lung is a mannequin-type lung…”

Authors must include an image showing the Mr. Lung.

Comment

Page 5. Line 120. “(Figure 1)”.

The number of participants in each group is a different number (Group Lecture N = 32 vs Group Lecture + Visualizing N = 39). It seems that there is no statistical reason to divide the participants in this no equal division. The problem of this situation comes from the fact that there is not a sample size calculation. The no equal distribution is not helping for getting reliable results.

Comment

Page 6. Line 126. “The students in the visualization group”

A) In the Introduction section, the authors say that "we created a simple graphical and mental visualization of lung sounds using a combination of lines and circles in lung auscultation". Authors must explain the procedure for the creation of that graphical tool. It was based on some other graphical images. Was it created by consensus? Did the authors perform any pilot study or experiment to explore if the graphics were easy to understand? The graphics showing the respiratory sounds are the main tool that the authors are using for the teaching-learning, hence, that tool must have a strong scientific background. While this is a nice tool, the authors must detail the origin of this tool.

B) Authors must explain in detail the additional training. For instance, did the students watch the graphics in a computer? With a Power Point Presentation? Or in a printed version? How long does the training take? It was a single session? How many times the students drew the representation of the lung sounds? Was it made by hands. What are the indications to mentally visualize lung sounds? Because this is a main section and a main part of the study, the learning with the graphical representation of the lung sounds must be well detailed. Otherwise, the study is not replicable. This teaching method should be tested before the method was applied for a study; the limited description of the method only causes doubts.

Comment

Page 7. Line 164. “Likert Scale”.

A Likert Scale questionnaire needs of a validation, Cronbach’s alpha must be included to know the reliability of the questionnaire. Hence, this evaluation tool lacks the minimum characteristics of validation.

Results section

Comment

Page 10. Line 209. “Finally, 65 students…”

If authors had a sample size calculation, one could know if this 65 participants is a suitable number for the study.

Comment

Page 10. Line 217. “control”

Authors must use same identification name for groups, sometimes they say "lecture group", other times they say "control group". Use the same ID in all the manuscript and the reader will not get confused.

Page 11. Line 228. “compared to that of the control group”.

Ok, but there were no significant statistical differences. Statistical tests are helpful to accept or reject the hypothesis. When the hypothesis is rejected, is fine, there is no need to give positive interpretation in something that disagree with your hypothesis.

Discussion section

General comment

The discussion section describes two main findings:

“The visualization of lung sounds can improve medical students” confidence in their lung auscultation.”

“The visualization of lung sounds may improve medical students’ ability to auscultate lung sounds, although the effect is limited.”

The first finding must be explained by the authors in a clear way. This was a finding that support the use of the graphics that represents the lung sound to gain confidence. The authors must explain what learning process occurred in the medical students that gave confidence in their lung auscultation. How can that have happened? It was an effect caused only by the visualization. What mental process might occur? Is there any positive link between the visualization and the training with the simulation model? Does the confidence in their lung auscultation have an impact on the ability to auscultate lung sounds? Some explanation is given in the discussion, but the explanation is more a description of findings from other authors.

The second fining also needs an explanation. It seems that there is no relation between the confidence and the performance on clinics by the medical students. How the authors explain that the visualization of lung sounds had a limited effect on the students’ learning? Maybe the graphics by themselves were the problem. Do the authors know if the graphics were easy to understand? Or if the learning protocol was well-designed to get a positive impact? Is there any relation between the learning with visualization and the learning with the simulation model.

Because the findings are not well explained in the discussion, it seems that the experimental design was not well designed to solve specific question. But more important, it seems that the experimental design was not well designed to explore some hypothesis. Hypothesis is also a guide to understand or to explain the results and evidence. Hence, the discussion must improve to show the scientific value of the study.

6. PLOS authors have the option to publish the peer review history of their article (what does this mean?). If published, this will include your full peer review and any attached files.

Reviewer #1: No

Reviewer #2: No

Reviewer #3: **Yes: **Bernardino Cerda

---

## [Author Response · Author response to Decision Letter 0]

25 Nov 2022

November 24, 2022

Ayse Hilal Bati, MD, PhD

Academic Editor 

PLOS ONE

Dear Dr. Ayse

Ref. No.: PONE-D-22-25852

We wish to re-submit the manuscript titled, “Does simple graphical and mental visualization of lung sounds improve the auscultation skills of clinical clerkship students? (Revised title: The effects of simple graphical and mental visualization of lung sounds in teaching lung auscultation during clinical clerkship: a preliminary study)”

We have revised our manuscript accordingly and have provided a point-by-point response to the reviewers’ comments. This is attached herewith. The changes to the manuscript are shown in red font.

We believe that our revised manuscript has suitably incorporated the reviewers’ suggestions and is significantly improved over our initial submission. We trust that it is now suitable for publication in PLOS ONE.

Moreover, Dr. Tajima contributed to the interpretation of the data and the collection of some of the original data in the revision of this manuscript. Therefore, he is added as a new co-author. 

Thank you for your consideration. I look forward to hearing from you.

Sincerely,

Hajime Kasai, MD

Department of Respirology, Graduate School of Medicine

Chiba University, 1-8-1 Inohana, Chuou-ku Chiba 260-8670, Japan

Telephone: 81-43-222-7171 Ext.71014

Fax: 81-43-226-2176

E-mail: daikasai6075@yahoo.co.jp  

Response to Reviewer #1’s comments

General Comment:

Thank you for submitting this manuscript. The following are my questions/comments for the authors to address prior to resubmitting and another round of peer review:

I said I do not know if statistics were performed adequately because there is not enough detail in the statistical methods for me to determine if the tests were the correct ones. See info below about statistical analysis.

Data availability: you shared your analysis but not your data. please upload de-identified data to an online data repository to fulfill this requirement.

Response: 

We express our strong appreciation for your insightful comments on our manuscript. The comments have helped us to improve the manuscript significantly. We have marked the relevant changes in red so you can easily identify them.

The data analyzed in this study have been attached as an Excel file without identifiable personal information.

Comment #1: Lines 40-42: the p value is not significant, nor even close. Therefore, you can’t say that there was any difference between groups in score 

Response: 

Thank you for this comment.

As the reviewer pointed out, there was no significant difference between the two groups’ test results regarding lung auscultation.

These results suggest that our educational method can be insufficient to improve participants’ performance. Therefore, in the future, we will develop and validate an educational method with further improvements for the visualization of lung sounds.

According to the comment, we have reworded the related sentence as follows:

(p2. Result section in Abstract.)

Original: “Although there were no differences between the pre- and post-tests scores of both groups, the score of the visualization group tended to increase compared with that of the control group (p=0.623).”

Revised: “However, there were no differences between the pre-and post-tests scores of both groups (p=0.623).”

Comment #2: Line 49: suggest adding simulation, model

Response: 

Following the comment, we have reworded the related sentence as follows:

(p2. Result section in Abstract.)

Original: “Keywords (user): auscultation; lung sounds; visualization”

Revised: “Keywords (user): auscultation; lung sounds; model; simulation; visualization”

Comment #3: Lines 56-59: I don’t feel like these 2 sentences are relevant to your paper and would remove them from your intro. Possibly you could work them into your discussion? But because you used med students, it doesn’t seem important how physicians and pulmonologists score relative to med students.

Response: 

Thank you for this insightful feedback.

These references are cited to demonstrate the need for better education on lung auscultation during medical school, as physicians’ auscultation skills are reported to have not improved since medical school. We also believe that it is important for learners to improve their skills in lung auscultation continuously. Accordingly, we added the following text to convey our intentions:

(p3. 1st paragraph of Introduction)

Added: “Therefore, medical students need more effective education programs on lung auscultation to improve their skills continuously.”

Comment #4: Lines 63-64: reference 6 refers to one simulator. Do you have references for more simulators, for textbook training, and for other audio sources?

Response: 

Thank you for pointing this out.

I have added additional references regarding simulators and audio sources for lung auscultation, not all of which are research articles. I have also added several texts, including Japanese sources, as references.

Accordingly, we have revised the related sentences as follows: 

(p3. 2nd paragraph of Introduction)

Original: “Recent advances in digital stethoscopes have made it possible to record lung sounds and share them among multiple medical professionals, as well as to analyze lung sounds automatically, perform auscultation wirelessly, and provide telemedicine [5]. However, effective education for lung auscultation has not been developed. The skills of lung auscultation have been taught using simulators in addition to textbooks and audio sources [6]. Moreover, in recent years, it has become possible to listen to lung sounds through internet video-sharing sites and smartphone applications [7–9].”

Revised: “Recent advances in digital stethoscopes have made it possible to record lung sounds and share them among multiple medical professionals, as well as to analyze lung sounds automatically, perform auscultation wirelessly, and provide telemedicine [5-7]. However, effective education for lung auscultation has not been developed. The skills of lung auscultation have been taught using simulators in addition to textbooks and audio sources [8-12]. These simulators can improve learner confidence and knowledge of lung sound auscultation [8,9,11]. Moreover, in recent years, it has become possible to listen to lung sounds through internet video-sharing sites. At the same time, only a limited number of materials can serve as appropriate teaching materials [13]. In addition, web-based materials and computer applications on pulmonary auscultation are available [14-17].”

5. Kim Y, Hyon Y, Lee S, Woo SD, Ha T, Chung C. The coming era of a new auscultation system for analyzing respiratory sounds. BMC Pulm Med. 2022;22: 119.

6. Park DE, Watson NL, Focht C, et al. Digitally recorded and remotely classified lung auscultation compared with conventional stethoscope classifications among children aged 1-59 months enrolled in the Pneumonia Etiology Research for Child Health (PERCH) case-control study. BMJ Open Respir Res. 2022;9.

7. Lapteva EA, Kharevich ON, Khatsko VV, et al. Automated lung sound analysis using the LungPass platform: a sensitive and specific tool for identifying lower respiratory tract involvement in COVID-19. Eur Respir J. 2021;58.

8. Kaminsky J, Bianchi R, Eisner S, et al. Respiratory Auscultation Lab Using a Cardiopulmonary Auscultation Simulation Manikin. MedEdPORTAL 2021;17: 11107.

9. Arimura Y, Komatsu H, Yanagi S, et al. [Educational usefulness of lung auscultation training with an auscultation simulator]. Nihon Kokyuki Gakkai Zasshi 2011;49: 413-8.

10. Bernardi S, Giudici F, Leone MF, et al. A prospective study on the efficacy of patient simulation in heart and lung auscultation. BMC Med Educ. 2019;19: 275.

11. Yoshii C, Anzai T, Yatera K, Kawajiri T, Nakashima Y, Kido M. [A new medical education using a lung sound auscultation simulator called “Mr. Lung”]. J UOEH 2002;24: 249-55.

12. Ward JJ, Wattier BA. Technology for enhancing chest auscultation in clinical simulation. Respir Care 2011;56: 834-45.

13. Sunderland N, Camm CF, Glover K, Watts A, Warwick G. A quality assessment of respiratory auscultation material on YouTube. Clin Med. (Lond) 2014;14: 391-5.

14. Higashiyama S, Tamakoshi K, Yamauchi T. Effectiveness of a new interactive web teaching material for improving lung auscultation skills: randomized controlled trial for clinical nurses. Nagoya J Med Sci. 2022;84: 526-38.

15. Cugell D, Gavriely N, Zellner D. Lung Sounds and the Stethoscope — a Lung Sounds Primer. MedEdPORTAL 2012;8: 9066.

16. Kraman S. Lung Sounds: An Introduction to the Interpretation of Auscultatory Findings. MedEdPORTAL 2007;3: 129.

17. Malmartel A, Ecollan M, Bories MC, Jablon E, Planquette B, Ranque B. [Evaluation of the use of a simulation software in the learning of cardiopulmonary auscultation in undergraduate medical students]. Rev Med Interne 2020;41: 653-60.

Comment #5: Line 88 - suggest changing grade to cohort. This is a better term to be internationally understood.

Response: 

Thank you for the useful suggestion.

Accordingly, we combined the description of the clinical clerkship (CC) in the Department of Respiratory Medicine with the preceding CC paragraph and changed the related sentences:

(p. 7, 1st and 2nd paragraph of study samples in Material and Methods)

Original: “CC in the Respiratory Medicine department and study samples

In our department’s CC, groups of seven to eight medical students underwent a four-week training program as members of a medical team of doctors and residents. All medical students were assigned two to four patients during this four-week period and performed daily physical examinations, including auscultation of the lungs.

A total of 89 medical students underwent CC in Respiratory Medicine at Chiba University Hospital between December 2020 and November 2021.

Revised: “Study design and samples

“This single-center study included a cohort of 89 medical students who underwent CC in Respiratory Medicine at Chiba University Hospital between December 2020 and November 2021.” 

Comment #6: Lines 91-93: can you clarify how the lecture included a simulator? did the teacher use the simulator and the students watched? did the students use the simulator? if so, were they in groups or as individuals?

Response: 

The purpose of the lecture was to practice auscultation techniques and understand the mechanisms behind lung sounds. Therefore, the simulator was used to practice listening to various lung sounds with a stethoscope. Four simulators were available. About six students performed auscultation for one simulator to practice listening to various lung sounds with a stethoscope.

Accordingly, we added the following sentences:

(p. 6, 1st paragraph of setting in Material and Methods)

Added: “In the third year, before the CC, students attended one 60-minute lecture on lung auscultation using a simulator (Mr. Lung®, Kyoto Kagaku Co. Ltd., Kyoto, Japan) as preparation for CC. Mr. Lung® is a mannequin-type lung sounds auscultation simulator widely used in medical schools in Japan (S1A Fig) [9,11]. The purpose of the lecture was to practice lung auscultation and understand the mechanisms of lung sounds. Four simulators were available; about six students performed auscultation using one simulator to practice listening to various lung sounds with a stethoscope.”

Comment #7: Lines 118-120: this study has an A vs A+B study design. Meaning lecture vs lecture + visualization training. This is a weak study design because it doesn’t compare groups with equivalent time spent learning. We would expect students who spent more time in instruction (the visualization group) to be more confident and out-perform the lecture only group. A better study design is to compare 2 groups with equal time spent learning, and only the method of learning is different. Your study design with uneven educational time should be added to the study limitations.

Response: 

Thank you for raising this concern. 

As you highlighted, the fact that the amount of time spent teaching visualization of lung sounds differs between the groups is an important limitation of this study.

Accordingly, we added the following sentences:

(p19, Study limitations)

Added: “Second, time spent on lung auscultation education differs between the two groups with and without time spent on training to visualize lung sounds.”

Comment #8: Line 134 - “of the inspiratory”? I don’t understand this wording.

Response: 

Thank you for your query. According to the comment, we have reworded the related sentence as follows:

(p. 7, 4th paragraph of Procedure of education on lung sound auscultation with visualization in Material and Methods)

Original: “Coarse crackles are presented as circles which are drawn at a lower position on inspiratory and/or expiratory. Fine crackles are presented as circles that are drawn at the end of the inspiratory and are concentrated at a high position.”

Revised: “Thus, coarse crackles are presented as circles drawn at a lower position on inspiration and/or expiration. Fine crackles are presented as circles that are drawn at the end of the inspiration and are concentrated at a high position.”

Comment #9: Lines 135-136: this is repetitive

Comment #10: Lines 140- 152: this is repetitive and is too long for a figure caption. make the figure caption just the first sentence.

Response: 

Thank you for your comment.

The captions to each figure have been limited only to the title. The other captions have been moved to the end of the manuscript.

Comment #11: Lines 164-165: please define all steps of the Likert scale, not just 1 and 5.

Response:

Only indications of 1 and 5 were presented in the actual questionnaire to the students in this study. In addition, although not indicated on the questionnaire form, it was explained orally that 3 meant fair. However, decisions other than 1, 3 and 5 were left to the students themselves.

We will make sure to incorporate a more detailed description in the questionnaires for our future studies.

Accordingly, we have added the related sentence as follows:

(p9. 1st paragraph of Questionnaire in Materials and Methods)

Added: “Indications 1 and 5 were presented in the actual questionnaire to the students. In addition, although not indicated on the questionnaire form, it was explained orally that 3 meant fair.”

Since it is impossible to retroactively add the above indication that has not been provided in the actual questionnaire, no changes were made to the questionnaire described in the method. This limitation has been added to the revised manuscript.

(p19, Study limitations)

Original: “First, as it was performed at just one university in Japan, the number of students was small.” 

Revised: “First, in terms of research design, this was a single-site study with a small number of participants in an uncontrolled environment and relied partly on students’ self-assessments for data collection. Therefore, a larger number of participants based on the calculation of the required sample size and the use of a reliable questionnaire are required in the future.”

Comment #12: Lines 176-177: it would be good to remind the reader here how much time there was in between the lecture and the visualization training, and between that training and the performance test.

Response: 

Thank you for the useful suggestion.

The duration of the lecture on lung auscultation in CC was one hour, while additional 30 minutes were spent teaching the visualization of lung sound for the visualization groups. For pre-and post-test, four simulators were used, and two students performed lung auscultation simultaneously for one simulator. When the students answered the question, the next question was presented. The test took approximately 30 minutes.

Accordingly, we added the related sentences as follows:

(p. 7, 3rd paragraph of Procedure of education on lung sound auscultation with visualization in Material and Methods)

Original: “The students in the visualization group received additional training using a diagram of graphically visualized lung sounds, as shown in Figure 2.”

Revised: “The students in the visualization group received approximately 30 minutes of additional training using a diagram of graphically visualized lung sounds during the lecture on lung auscultation in CC, as shown in Fig 2 and S2 Fig.”

We have also added S2 Fig to provide details regarding the lecture on lung auscultation in CC and additional training on the visualization of lung sounds.

 

S2 Fig. Details of the lecture on lung auscultation conducted in CC and additional training on the visualization of lung sounds.

  

In addition, we revised figure 3 to show the duration of the lecture and the pre- and post-test as follows:

Revised Figure 3

Comment #13: Lines 197-198: please describe how you tested your data for normality and what the result was.

Response: 

Thank you for the insightful comment.

Normality was evaluated using the Shapiro-Wilk test in JMP Pro14. The results are shown in the following table. The results showed that the score of lung auscultation in pre-and post-tests for the control group was normally distributed, while the other parameters were non-normally distributed.

The difference in the distribution of the score of lung auscultation in pre-and post-tests between the two groups can be caused by the small sample of our study, as mentioned in the Limitation section. Further studies with larger sample sizes are planned in the future.

Table. P-value in the test of normality of each parameter (n=65).

 Visualization group (n = 35) Control group

(n = 30)

Satisfaction ＜0.001 ＜0.001

Confidence 

Pre-questionnaire ＜0.001 0.010

Post-questionnaire ＜0.001 ＜0.001

Score of lung auscultation 

Pre-test 0.008 0.610

Post-test 0.049 0.747

(p. 12, Statistical analysis in Material and Methods)

Added: “Normality was evaluated using the Shapiro-Wilk test.”

(p. 13, 1st paragraph in Result)

Added: “The pre-and post-tests for the control group were normally distributed, while the other parameters were non-normally distributed.”

Comment #14: Line 198: “compare the answers” - the answers to what?

Response: 

In this study, the Wilcoxon signed-rank test was used to compare the answers on the pre-and post-questionnaire about participants’ confidence in lung auscultation and the scores of the pre-and post-test with lung auscultation. Insufficient information was provided.

Accordingly, we modified the related sentences as follows:

(p. 12, Statistical analysis in Material and Methods)

Original: “The Wilcoxon signed-rank test was used to compare the answers before and after our education.”

Revised: “The Wilcoxon signed-rank test was used to compare the questionnaire answers about participants’ confidence in lung auscultation and the test scores of lung auscultation before and after the training.”

Comment #15: Lines 212-213: this p value is not significant, nor even close, so you cannot say that there was any difference between groups in satisfaction level.

Comment #16: Lines 227-229: this p value is not significant, nor even close, so you cannot say that there was any difference between groups.

Response: 

Thank you for the suggestion.

Accordingly, we have revised the sentences as follows: 

(p. 13, 1st paragraph of Questionnaire in Results)

Original: “Based on an analysis of their responses, students’ satisfaction level with our education was high in both groups, but slightly higher in the visualization group (visualization group: 4.5±0.1 vs. control group: 4.2±0.1, p = 0.150).”

Revised: “Based on an analysis of their responses, students’ satisfaction level with our education was high in both groups, but there was no significant difference (visualization group: 4.5±0.1 vs. control group: 4.2±0.1, p = 0.150).”

(p. 14, 1st paragraph of Results of tests of lung auscultation in Results)

Original: “Although there was no difference between the scores of the pre- and post-tests of both groups (pre-test, p = 0.827; post-test, p = 0.290), the visualization group scores tended to increase as compared to that of the control group (adjusted mean difference between pre- and post-tests, 3.7±0.5 vs. 3.1±0.6, p = 0.424).”

Revised: “There was no difference between the scores of the pre-and post-tests for both groups (pre-test, p = 0.827; post-test, p = 0.290). In addition, there was no change in scores between the two groups (adjusted mean difference between pre-and post-tests, 3.7±0.5 vs. 3.1±0.6, p = 0.424).”

Comment #17: Line numbering stopped at the discussion. Please correct this when resubmitting.

Response: 

We apologize for the oversight. The setting for line numbering was interrupted before and after Table 2. After the re-configuration of the setting, the revised manuscript included the line numbering even after the Discussion section.

Comment #18: Discussion paragraph 1, sentence 4: you cannot say this because there was no difference between groups.

Comment #19: Discussion paragraph 2, sentence 1: you can’t say that it may help their lung auscultation skills, because there was no difference between groups.

Response: 

Thank you for the suggestion.

As we responded in comments #16 and #17, there was no statistically significant difference. We have changed this description as follows: 

(p. 16, 1st paragraph of Discussion)

Original: “Second, the visualization of lung sounds may improve medical students’ ability to auscultate lung sounds, although the effect is limited.”

Revised: “Second, the visualization of lung sounds could not improve medical students’ ability to auscultate lung sounds; further improvements are needed.”

(p. 16, 2nd paragraph of Discussion)

Original: “The visualization of lung sounds in lung auscultation can boost medical students’ confidence in lung auscultation and help develop their lung auscultation skills.”

Revised: “The visualization of lung sounds in lung auscultation can boost medical students’ confidence in lung auscultation.”

Comment #20: Discussion paragraph 2: I would suggest having the sentence starting “Moreover,” start a new paragraph.

Response: 

Thank you for the suggestion.

Accordingly, we have split the paragraphs where indicated.

Comment #21: Discussion paragraph 2, sentence starting “since post-test scores were only...”: so how do you suggest that this training can be improved so that students achieve better scores? how can the lecture be improved? how can the visualization exercise be improved?

Response: 

The method to improve the effect of visualization of lung sounds cannot only rely on teaching students how to visualize in lectures but also on making them practice simple drawing and image-visualization of lung sounds repeatedly. As mentioned in the 4th paragraph of the Discussion, it may be necessary to repeatedly instruct students to visualize during hospital ward rounds with their supervisors and provide opportunities to draw visualizations about once a week. For effective learning, reflective observation and abstract conceptualization are necessary in addition to active experimentation and concrete experience. Repeating these four steps enhances learning.1 By visualizing lung sounds, students can learn not only whether the auscultated lung sound is appropriate or not but also how correct their judgment of the auscultated sound is. Each auscultation can be made more valuable with appropriate feedback from the instructor based on the lung sounds diagram students draw. This process may promote reflective observation and abstract conceptualization.

1. Kolb DA. Experiential learning: Experience as the source of learning and development: FT press, 2014.

In addition, the image of lung sounds can be different for each person, and it may be necessary to change the diagram to make it more intuitive.2 Therefore, breath sounds may be better shown in a diagram combining triangles rather than lines, as shown in the following figure. Furthermore, it may be easier to intuitively visualize the coarse and fine crackles by using an X instead of a circle as the symbol indicating the crackle. 

2. Scott G, Presswood EJ, Makubate B, Cross F. Lung sounds: how doctors draw crackles and wheeze. Postgrad Med J. 2013 Dec;89(1058): 693-7.

 

Figure. Proposed changes to lung sound visualization.

Accordingly, we have added the related sentences as follows. 

(p18-19. 5th and 6th paragraphs of Discussion)

Original: “Instruction to repeatedly visualize lung sounds may be insufficient. It is possible that retention by instructing the students to repeatedly visualize lung sounds during their daily practice may have been more effective. Additional opportunities to draw lung sounds using the simulator or recorded lung sounds may have enhanced the effectiveness of our educational method; this possibility, along with the aforementioned one, could be addressed by future research.”

Revised: “Only instruction to repeatedly visualize lung sounds may be insufficient. The study did not investigate how often students mentally imaged lung sounds during lung auscultation. Therefore, it is not clear how many students performed lung auscultation by visualizing lung sounds. Retention by instructing students to visualize lung sounds during their daily practice may have been more effective. Additional opportunities to draw lung sounds using the simulator or record lung sounds may have enhanced the effectiveness of our educational method. Furthermore, it may be necessary to instruct students to visualize lung sounds repeatedly during daily rounds with their supervisors and provide opportunities to draw visualizations at least once a week. 

Moreover, for effective learning, reflective observation and abstract conceptualization are necessary in addition to active experimentation and concrete experience. Repeating these four steps enhances learning [30]. Malmartel et al. examined the effect of combining a cardiopulmonary auscultation application (Medsounds™, Interactive Systems For Healthcare®) with clinical practice on medical students’ auscultation of heart and lung sounds [17]. Similar to our study, they used a high-fidelity simulator to assess the auscultation of coarse crackles in pneumonia, showing that students’ auscultation performance improved. In addition to the application, they suggested that appropriate supervision from the educator may have had an important impact. 

Another reason for the lack of a significant effect in our study can be that the diagram for the visualization of lung sounds was not appropriate for learners to image mentally. The image of lung sounds can be different for each person; thus, it may be necessary to change the diagram to make it more intuitive [31]. For example, breath sounds may be better shown in a diagram combining triangles rather than lines, as shown in the following figure. Furthermore, it may be easier to intuitively visualize the coarse and fine crackles using an X instead of a circle (S5 Fig). In this study, students in the visualization group drew lung sound diagrams only as an aid during the post-test. In the future, students will be provided with multiple opportunities to draw lung sounds and their lung sound diagrams will be evaluate.”

In addition, we added the above figure as S5 Fig.

Comment #22: Discussion paragraph 3, sentence 1: you can’t say this. the p value was not significant.

Response: 

Thank you for the suggestion.

As we responded in comments #16 and #17, there was no statistically significant difference. We have changed this description as follows: 

(p. 18, 5th paragraph of Discussion)

Original: “The effect on the improvement score of lung auscultation was limited. Although there was no significant difference between the post-test scores of both groups, the visualizing group’s scores tended to improve more than that of the control group.”

Revised: “This study did not find any significant effect of visualization on lung auscultation. There was no significant difference between the post-test scores of both groups, and there was no change in scores between the two groups. Since post-test scores were only around half the maximum number of points, confidence in lung auscultation will not suffice.”

Comment #23: Discussion paragraph 3, sentence 3: this needs to be highlighted as a limitation of the study. if the study was to evaluate non-visual training (lecture only) vs non-visual training (lecture) and visual training, then the non-visual portion should not have included visual training! this muddies the waters.

Response: 

We thank the reviewer for raising this concern. 

In our study, only the visualization group was taught how to visualize lung sounds in the lecture. The control group was given only lectures, while the visualization group was taught about visualization of lung sounds between lectures.

These statements were confusing. Therefore, we have changed the related sentences as follows:

(p. 18, 5th paragraph of Discussion)

Original: “During the lecture, we presented a visualization of lung sounds and asked the students to draw a lung sound which they heard during the lecture.”

Revised: “For the visualization group, we presented a visualization of lung sounds and asked students to draw a diagram of the lung sound that they heard during the lecture.”

Comment #24: Discussion paragraph 3, sentence starting “Furthermore...”: this sentence does not make sense. please revise.

Response: 

In daily CC, students are often asked to describe the lung sounds they heard from patients. However, it is unclear how students listen to lung sounds in patients and how they evaluate these sounds. We believe that a more detailed understanding of the student’s auscultation process by the supervisor will lead to more effective education.

Accordingly, the following revisions have been made to convey the above.

(p. 17, 4th paragraph of Discussion)

Original: “Furthermore, it is not clear how the medical students actually heard the lung sounds from their answers that a lung sound abnormality was heard.”

Revised: “Furthermore, it is unclear how students listen to lung sounds and how they evaluate them in real patients.”

Comment #25: Study limitations: you need to add the A vs A+B design, unequal time spent in teaching in the 2 groups, and that the lecture only group also got a discussion of visualization during the lecture

Response: 

Thank you for raising this concern. 

As we responded in comment #7, we added the description of the unequal time spent teaching in the two groups as a study limitation. 

Additionally, the control group did not discuss the visualization of lung sounds in the lecture in this study. We have corrected this point, as shown in response to comment #23

Comment #26: Conclusions: sentence 1 - you cannot say that it may improve their auscultation skills - your p value was not significant.

Response: 

Thank you for the suggestion.

Accordingly, we have revised the related sentence as follows:

(p. 20, Conclusion)

Original: “Although the effect of lung sound visualization is limited, it can be undertaken without additional devices such as special simulators and serve as a useful educational strategy.”

Revised: “Although the visualization of lung sounds did not affect significantly lung auscultatory ability to identify lung sounds, its advantages include the possibility of being undertaken without additional devices, such as special simulators, thus serving as a useful educational strategy.”

Comment #27: Seneral word on statistics - you seem to want to say that there are increases when the p value is insignificant. this is not acceptable. if you want to calculate effect size for the variables you measured, then you could say that the p value was not significant but the effect size was small/med/large and discuss implications - e.g. do you need a larger study to detect a different p value? Please consider adding effect sizes to your calculations. If you do not, all you can say is that there was no difference in any measure except confidence, because the p values are insignificant.

Response: 

As pointed out in comments #1, 15, 16, 18, 19, 22, and 26, we have changed the description of the parameters that did not show a significant difference between the visualization and control groups. 

In addition to the small sample size of this study, further improvement of the educational method is needed to show more effects. It may have been necessary to examine various effects other than confidence and auscultation scores of students, such as the degree of understanding of lung sounds and identifying abnormalities.

We plan to conduct further research to evaluate the effect of the modified educational methods using the visualization of lung sounds based on the reviewer’s comments. Thus, we have expanded our discussion in this revised version of the manuscript, as shown in the above responses; this study can serve as a basis for future studies. Therefore, we regard this study as preliminary and are planning to conduct a new study with improvements to address the problems of this study. 

We want to express our gratitude to the reviewers once again.

Comment #28: References 7-8: can you find more scientific references (e.g. journal articles) rather than citing sources such as youtube?

Response: 

Thank you for raising this concern. 

As you pointed out, it is not scientific to include YouTube as a reference. There have been no studies using video-sharing sites such as YouTube for teaching lung auscultation. At the same time, we find an article that evaluates whether they are an appropriate teaching tool for lung sounds. 

Accordingly, we have revised the related sentence and changed the reference as follows:

(p3. 2nd paragraph of Introduction)

Original: “Moreover, in recent years, it has become possible to listen to lung sounds through internet video-sharing sites and smartphone applications [7–9].”

Revised: “Moreover, in recent years, it has become possible to listen to lung sounds through internet video-sharing sites. At the same time, only a limited number of materials can serve as appropriate teaching materials [13].”

13. Sunderland N, Camm CF, Glover K, Watts A, Warwick G. A quality assessment of respiratory auscultation material on YouTube. Clin Med (Lond) 2014;14: 391-5.

 

Response to Reviewer #2’s comments

Comment: Kasai H et al present here the results of a study evaluating an interesting teaching method. Indeed, pulmonary auscultation is a fundamental clinical skill for any physician. The educational tool studied here also has the advantage of being simple and inexpensive.

The article is very well written, concise and presents the methodology (the pedagogical concept) and the results quite clearly.

Several comments and minor revisions should be made to the manuscript.

Response: 

We appreciate your comments. 

We have marked the respective changes with underlined text to make them easily identifiable.

Introduction

Comment #1: another medical university evaluated the interest of simulation software in the learning of cardiopulmonary and published their results (PMID 32660857). The reference could be cited because evaluation of auscultation skill was done using a high-fidelity simulator after training students with a specific software.

Response: 

Thank you for your suggestion and for presenting this useful report. 

We have used the article in the background as one of the educational practices of cardiopulmonary physical examination using the application with other articles. 

Accordingly, we have revised the related sentence as follows:

(p4. 2nd paragraph of Introduction)

Original: “Moreover, in recent years, it has become possible to listen to lung sounds through internet video-sharing sites and smartphone applications [7–9].”

Revised: “In addition, web-based materials and computer applications on pulmonary auscultation are available [14-17].”

17. Malmartel A, Ecollan M, Bories MC, Jablon E, Planquette B, Ranque B. [Evaluation of the use of a simulation software in the learning of cardiopulmonary auscultation in undergraduate medical students]. Rev Med Interne 2020;41: 653-60.

Similar to our study, they used a high-fidelity simulator to assess the auscultation of coarse crackles in pneumonia, and the auscultation performance of students improved. In addition to a cardiopulmonary auscultation application (Medsounds™, Interactive Systems For Healthcare®), they suggested that appropriate supervision from the educator may have had an important impact. Thus, the limited role of the educator in our study might have reduced the effect of the training. For these reasons, we have included this successful case study in our discussion as one of the references that can improve our educational methods.

(p18. 5th paragraph of Discussion)

Added: “Malmartel et al. examined the effect of combining a cardiopulmonary auscultation application (Medsounds™, Interactive Systems For Healthcare®) with clinical practice on medical students’ auscultation of heart and lung sounds [17]. Similar to our study, they used a high-fidelity simulator to assess the auscultation of coarse crackles in pneumonia, showing that students’ auscultation performance improved. In addition to the application, they suggested that appropriate supervision from the educator may have had an important impact.”

Methods/results sections:

Comment #2: The 3 questions asked to students (P7) might be presented in a table in the method section; results for each question might follow the same order in the result section (p10). Please change the order of questions or the order os answers and recall the question by putting it in brackets in the results section for each of the sentences detailing the answers. Reading and understanding of results will be facilitated.

Response: 

Thank you for your suggestion. We have compiled the questions into the following table.

Table 1. Questionnaire items before and after the training on lung auscultation.

Before the CC in Respiratory Medicine Responses

(A1) How is your current lung sound auscultation ability? Five-point Likert scale

1 (Not confident [cannot listen at all]) to 5 (Confident [able to listen and distinguish between lung sounds and rales])

(A2) Have you ever heard each lung sound from a patient? 

Decreased respiratory sounds, bronchial breathing, prolonged expiration, coarse crackles, fine crackles, wheezes, rhonchi, squawk, and pleural friction rub Yes or No

(A3) How confident are you in each lung sounds?

Same items as in question A2. The response options are same as in question A1.

After the CC in Respiratory Medicine 

(B1) How is your current lung sound auscultation ability? The response options are same as in question A1.

(B2) Have you ever heard each lung sound from a patient?

Same items as in question A2. The response options are same as in question A2.

(B3) How confident are you in each lung sounds?

Same items as in question A2. The response options are same as in question A1.

(B4) What is your level of satisfaction with the education program on lung auscultation? Five-point Likert scale

1 (extremely poor) to 9 (extremely good)

Accordingly, we have added the related sentences.

(p. 13, 1st and 2nd paragraph of Questionnaire in Results)

Added: “(A2 and B2 Have you ever heard each lung sound from a patient?)”

Added: “(A1 and B1: How is your current lung sound auscultation ability?)”

Added: “(A3 and B3 How confident are you in each of the lung sounds?)”

Comment #3: In the method section page 8, table 1 appears in the text while it might appear with tables. Same comment for tittle and footprints for Fig2.

Response: 

We included the caption and the title of Figure 2 in the text of the first manuscript. Only the figure titles are reported in the manuscript, whereas the extended captions have been moved to the end of the manuscript.

Comment #4: In the method section “performance of lung auscultation” details that “multiple-choice form that included a space to draw a figure where there was an abormality”. No results concerning the drawings proposed by the students during this post-test evaluation have been reported in the results section of the manuscript. Are they unexploited data or unexploitable data? Can you clarify this point, in particular how can we compare the graphic representations during the post-test evaluation with those made during the lecture?

Response: 

The diagrams on the response form were not evaluated in this study, but they were used to support the students in answering each question. In the future, we will provide students with multiple opportunities to draw lung sounds and will evaluate how they draw lung sound diagrams.

Following the reviewer’s comment, we have revised the text as follows:

(p. 11, 1st paragraph of Performance of lung auscultation in Materials and Methods)

Added: “Moreover, students used the diagrams on the answer sheet as an aid in answering the questions. The diagrams students drew were not included in the evaluation.”

(p. 19, 5th paragraph of Discussion)

Added: “In this study, students in the visualization group drew lung sound diagrams only as an aid during the post-test. In the future, students will be provided with multiple opportunities to draw lung sounds and their lung sound diagrams will be evaluate.”

Comment #5: Method section: Did the students have to follow a particular examination plan (number and order of auscultatory foci/number of respiratory cycles per focus)?

I suggest presenting in a simple table (or in the text) some data (if available) concerning the students of each group: age, sex ratio, results of the faculty exams and grades/evaluations of the clinical clerkship.

Response: 

Thank you for your valuable comment.

In this study, students were not limited in time and number for lung auscultation while answering each question in pre-and post-tests.

Accordingly, we added the related sentence as follows:

(p. 11, 1st paragraph of Performance of lung auscultation in Materials and Methods)

Added: “Students were not limited in time and number for lung auscultation while answering each question in pre-and post-tests.”

Table A shows the age, sex, and results of CC in Respiratory Medicine in two groups. The results of CC in Respiratory Medicine were calculated based on the performance of medical interviews and physical examination during CC, presentation, contents of medical records, and summaries, except for the results of the lung auscultation test in this study. There were no significant differences in any of these parameters between the two groups.

 

Table. A Baseline characteristics (n = 65)

Parameter Visualization group

(n=35) Control group

(n=30) P-value

Age, years 23.2±1.6 22.6±0.8 0.123

Sex (Male / Female) 27/8 24/6 0.780

Result of CC in Respiratory Medicine 76.7±3.9 77.1±5.1 0.802

Data are presented as mean ± standard deviation.

Accordingly, we revised the related sentence as follows:

(p. 12-13, 1st paragraph of Results)

Original: “Finally, 65 students were included in the study.”

Revised: “Finally, 65 students were included in the study. The visualization group included 35 students (age 23.2±1.6 years; male/female 27/8), and the control group included 30 students (22.6±0.8 years; male/female 24/6) (Fig 1). There were no significant differences in the age and sex ratio (p=0.123 and p=0.780, respectively). In addition, there was no significant difference in the results of CC in Respiratory Medicine, which did not include the results of the lung auscultation test between the two groups (maximum score: 25, 76.7±3.9 vs. 77.1±5.1, p=0.802).”

Discussion section

Comment #6: P 13, paragraph 1 The authors cannot conclude that “the visualization of lung sounds may improve student’s ability” based on their results. It seems necessary to rephrase this conclusion.

Response: 

As the reviewer pointed out, there was no statistically significant difference. We changed this description as follows: 

(p. 16, 1st paragraph of Discussion)

Original: “Second, the visualization of lung sounds may improve medical students’ ability to auscultate lung sounds, although the effect is limited.”

Revised: “Second, the visualization of lung sounds could not improve medical students’ ability to auscultate lung sounds; further improvements are needed.”

Comment #7: P13, paragraph 2 “Although the primary objective was to identify...”. This formulation is very confusing while identifying abnormal lung sounds was not the primary objective of the study. I understand that, the main objective of the auscultation skill is to be able to distinguish abnormal sounds from normal auscultation.

Response: 

Thank you for raising this concern. 

As you pointed out, lung auscultation’s primary objective is to discriminate between normal and abnormal auscultatory sounds, and the next step is to analyze abnormal sounds. 

Although recognizing the presence of abnormal lung sounds is certainly important, in actual clinical practice, identifying the feature of abnormal lung sounds can be necessary for clinical reasoning and determining plans depending on the patient’s condition. This study aimed to improve the ability to identify abnormal lung sounds. 

There was confusion between the general objective of lung auscultation and the research objective of our study.

Following the reviewer’s comment, we have revised the text as follows:

(p16. 2nd paragraph of Discussion)

Original: “Although the primary objective was to identify the lung sound abnormalities that were present in the patient, limited interest has been shown in determining the timing of breathing and/or the pitch of the abnormal sounds that can be heard.”

Revised: “Although the primary objective was to distinguish abnormal lung sounds from normal lung auscultation in the patient, limited interest has been shown in determining the timing of breathing and/or the pitch of the abnormal sounds that can be heard.”

Comment #8: p13, second last line: I suggest to start a new paragraph withe the sentence “Moroever....”

Response: 

Thank you for the suggestion.　 

Accordingly, we have split the paragraphs where indicated.

Comment #9: p14, the sentence “Since post-test scores were only...” could be moved to the beginning of the next paragraph.

Response: 

Following the reviewer’s comment, we have moved the sentence to the beginning of the next paragraph. 　 

Comment #10: p14, can the authors argue the sentence “However, the satisfaction level was also significantly higher in the visualization group, suggesting that visualization may have facilitated student’s better understanding of lung sounds”? To my knowledge, student’s satisfaction with education/lecture/course is associated with a better understanding. References to support authors point of view are required.

Response:

Thank you for the useful suggestion.

Although both satisfaction and learning effectiveness were often improved in various medical education studies, the correlation between satisfaction and learning effectiveness was not evaluated. As you pointed out, students’ satisfaction with education/lecture/course can be associated with a better understanding. However, through video clips, a study of education for cardiac auscultation was conducted, and a discrepancy between satisfaction and improvement in the performance of cardiac auscultation was found.1

Therefore, it is possible that satisfaction with education for auscultation is not directly associated with short-term improvement in auscultation ability. We need to verify the long-term improvement in the ability due to changes in learning attitudes.

1. Kagaya Y, Tabata M, Arata Y, Kameoka J, Ishii S. Employment of color Doppler echocardiographic video clips in a cardiac auscultation class with a cardiology patient simulator: discrepancy between students’ satisfaction and learning. BMC Med Educ. 2021 Dec 6;21(1): 600.

However, there was no significant difference in satisfaction levels between the two groups (visualization group: 4.5±0.71 vs. control group: 4.2±0.19, p = 0.150)), even though it appeared to be slightly higher in the visualization group. Thus, the sentence in the Discussion section included an error, as pointed out by other reviewers. Therefore, it has been removed.

As mentioned above, we consider the satisfaction level an important parameter to assess the results of educational methods, and we will continue to use it in our future research.

Comment #11: p14, second paragraph: “However, it was clear how often the student continued to visualize lung sounds in their daily practice” should be moved to the limit paragraph (p15).

Response: 

Thank you for the suggestion. 

As you pointed out, this is an important limitation.

Accordingly, we moved the following sentences:

(p19, Study Limitations)

Added: “Third, it is unclear how often students continued to visualize lung sounds in their daily practice.”

Comment #12: p14 : the last sentence “Furthermore” is not understable, please rephrase.

Response: 

Regarding the sentence you pointed out, I apologize for the misspelling of “Furthermore.”

In addition to the comment from Reviewer #1, we have revised the related sentence as follows:

(p. 17, 4th paragraph of Discussion)

Original: “Furthermore, it is not clear how the medical students actually heard the lung sounds from their answers that a lung sound abnormality was heard.”

Revised: “Furthermore, it is unclear how students listen to lung sounds and how they evaluate them in real patients.”

Comment #13: p15: the second limit discussed here is for me one strength: in respiratory departments, students have access to a wide range of abnormal lung sounds. This is well shown par figure S3.

Response: 

Thank you for the suggestion. 

As you pointed out, the Department of Respiratory Medicine can provide more opportunities to practice lung auscultation than other departments. As shown in S2 Fig, there was no difference in the number of experiences between the two groups. Therefore, we have removed the second limitation from the manuscript. 

Response to Reviewer #3’s comments

Comment: I thank you for the invitation to review the manuscript “Does simple graphical and mental visualization of lung sounds improve the auscultation skills of clinical clerkship students?” The manuscript describes a study that explored the impact of graphical and mental visualization of lung sounds on the confidence and performance of lung auscultation in medical students. The study explored an interesting research problem that has not been explored in the past. The authors had a good idea of using graphical and mental visualization of lung sounds for learning of lung auscultation. In addition, the learning of the lung sounds was supported by a learning with a simulation model. However, the study has many weaknesses that impact negatively on its scientific quality.

Response: 

We appreciate your encouraging comments about our study. We have marked the changes to the manuscript in response to your comments with yellow highlights to make them easily identifiable.

Title 

Comment #1: The title must be written like a statement but not like a question. A question does not show the impact or the value of the study.

Response: 

We apologize for the inappropriate title, which presented a question. We have rephrased the title as a statement and included the concept of the study. In addition, we are planning a new study to verify the effectiveness of lung sound visualization based on the valuable comments of the reviewers to this study. We have also designated this study as preliminary.

Following the reviewer’s comment, we have revised the title as follows:

(p. 1, Title)

Original: “Does simple graphical and mental visualization of lung sounds improve the auscultation skills of clinical clerkship students?”

Revised: “The effects of simple graphical and mental visualization of lung sounds in teaching lung auscultation during clinical clerkship: A preliminary study.”

Abstract Methods

Comment #2: The method section says that 79 students participated in the study. Hence, 35 students were part of the visualization group. But also, the method section says that 30 students were part of the control group. This is 65 students, but not 79 students. Authors must check this situation. Authors should include the name of the statistical test that they applied to perform the analysis of the data.

Response: 

The reviewer’s comment is correct. To clarify, the total number of students in this study was changed from 79 to 65 in the abstract.

In addition, we have added a description regarding the statistical test.

(p2. Methods of Abstract) 

Added: “The Wilcoxon signed-rank test and analysis of covariance were used to compare the answers to the questionnaire about confidence in lung auscultation and the scores of the lung auscultation tests before and after the training.”

Introduction section

Comment #3: Page 3. Line 56. Please check the sentence. Authors say that “in a study”, but at the end of the sentence they included two references. Therefore, is not “in a study”, but “in studies”.

Response: 

Following the reviewer’s comment, we have revised the related sentence as follows.

(p4. 1st paragraph in Introduction) 

Original: “In a study comparing the auscultation skills of medical students and physicians, there was no difference in skill, except for pulmonologists [3,4].”

Revised: “In studies comparing the auscultation skills of medical students and physicians, there was no difference in skill, except for pulmonologists [3,4].”

Comment #4: Page 3. Line 68. Bernardi et al… 

In the abstract, the authors mention that they used a simulator (Mr. Lung), but in the Introduction section they did not mention something about that. Authors must include a brief description of the simulator and its possible impact on the learning of the students. In the Bernardi et al. study, do those authors also use Mr. Lung? Or other simulation model? In the present study, what is the relevance of using the simulation model? Authors should explain that in the Introduction section.

Response: 

A mannequin simulator was utilized in the lectures of our usual education program because it has been verified to have a learning effect.1-4 Some studies used an audio player to play lung sounds for students’ evaluation and diagnosis.5-8 In contrast, other studies, including Bernardi et al., used mannequin simulators for lung sound. 

In this study, the simulator was used in the pre-and post-test to acquire clinically relevant auscultation skills with an awareness of the location and timing of breathing rather than simply playing sounds. In addition, we also aimed to verify whether the visualization of lung sounds is possible while performing the auscultation procedure with a stethoscope. 

1. Kaminsky J, Bianchi R, Eisner S, et al. Respiratory Auscultation Lab Using a Cardiopulmonary Auscultation Simulation Manikin. MedEdPORTAL 2021; 17:11107.

2. Malmartel A, Ecollan M, Bories MC, Jablon E, Planquette B, Ranque B [Evaluation of the use of a simulation software in the learning of cardiopulmonary auscultation in undergraduate medical students]. Rev Med Interne 2020; 41:653-60.

3. Yoshii C, Anzai T, Yatera K, Kawajiri T, Nakashima Y, Kido M [A new medical education using a lung sound auscultation simulator called "Mr. Lung"]. J UOEH 2002; 24:249-55.

4.　 Arimura Y, Komatsu H, Yanagi S, et al. [Educational usefulness of lung auscultation training with an auscultation simulator]. Nihon Kokyuki Gakkai Zasshi 2011; 49:413-8. 

5. Aviles-Solis JC, Storvoll I, Vanbelle S, Melbye H The use of spectrograms improves the classification of wheezes and crackles in an educational setting. Sci Rep 2020; 10:8461.

6. Andres E, Gass R Stethoscope: A Still-Relevant Tool and Medical Companion. Am J Med 2016; 129:e37-8.

7. Nguyen DQ, Patenaude JV, Gagnon R, Deligne B, Bouthillier I Simulation-based multiple-choice test assessment of clinical competence for large groups of medical students: a comparison of auscultation sound identification either with or without clinical context. Can Med Educ J 2015; 6:e4-e13.

8. Bohadana A, Izbicki G, Kraman SS Fundamentals of lung auscultation. N Engl J Med 2014; 370:744-51.

Accordingly, we have added the related sentences as follows:

(p10. 1st paragraph of Performance of lung auscultation in Materials and Methods)

Added: “The simulator was used to evaluate the relevant auscultation skills with an awareness of the location and timing of breathing rather than simply playing sounds.”

Bernardi et al. used the Kyoto-Kagaku patient simulator (Cardiology patient simulator “K Plus” training system, Model #11257–159, Kyoto Kagaku Co. Ltd., Kyoto, Japan, https://www.kyotokagaku.com/en/products_introduction/mw10/). Although this simulator was a different model from the simulator used in our study (Mr. Lung®), it was manufactured by the same company, Kyoto Kagaku, and it is similar in terms of the sounds that can be heard.

Accordingly, we have revised the related sentences as follows:

(p4-5. 2nd paragraph of Introduction)

Original: “Bernardi et al. examined the effects of simulator-based education on lung sounds and heart sounds and observed an improvement in the ability of auscultation of heart sounds, but no change in that of lung sounds [10].”

Revised: “Bernardi et al. examined the effects of simulator-based education on lung sounds and heart sounds (Cardiology patient simulator “K Plus” training system, Model #11257–159, Kyoto Kagaku Co. Ltd., Kyoto, Japan) and observed an improvement in the ability of auscultation of heart sounds, but no change in that of lung sounds [10].”

Comment #5: Page 4. Line 74. Is there any study exploring the learning of lung sounds by students with those spectrograms? If there is not, then authors can say that to give a better reason to perform its study. If there is some study, authors should mention it.

Response: 

Andrés et al. evaluated the effect of adding spectrogram information to listening for abnormal lung sounds and found that the spectrograms positively affected how medical students assigned a diagnosis with the help of lung sounds.1 Aviles-Solis et al. evaluated the effect of a spectrogram for the lung auscultation of medical students. They reported that using spectrograms was useful in identifying wheezes and crackles during lung auscultation.2 Although the descriptive schemas of lung sounds are used in various textbooks, the effectiveness of using visualizations of lung sounds other than spectrograms in actual auscultation has not been verified. In this study, the visualization had a limited effect on learners, but we believe its effectiveness can be verified with further improvements.

1. Andrès E, Gass R. Stethoscope: A Still-Relevant Tool and Medical Companion. Am J Med. 2016 May;129(5): e37-8.

2. Aviles-Solis JC, Storvoll I, Vanbelle S, Melbye H. The use of spectrograms improves the classification of wheezes and crackles in an educational setting. Sci Rep. 2020 May 21;10(1): 8461. 

(p5. 3rd paragraph of Introduction)

Original: “While it is useful to learn the characteristics of lung sounds [12], spectrograms are not available in all institutions, and it is difficult to visualize them with a personal stethoscope.”

Revised: “It is useful to learn the characteristics of lung sounds [20,21]; the use of spectrograms can be useful for medical students to diagnose lung sounds and identify abnormal lung sounds [21,22]. However, spectrograms may not be available in all institutions, and it is difficult to visualize them with a normal personal stethoscope.”

Comment #6: Page 4. Line 77. While it is useful…

Why the authors say that the spectrograms are not available in all institutions? Is there any evidence to support this statement? Maybe the statement should not be written as an absolute statement.

Response: 

Although there are some reports on the educational use of spectrograms, as shown in the Response to Comment #5, recording devices to obtain spectrograms are not widely used in general, including at our institution. However, there is no evidence of the prevalence of such devices.

Accordingly, we have revised the related sentences as follows:

(p5. 3rd paragraph of Introduction)

Original: “While it is useful to learn the characteristics of lung sounds [12], spectrograms are not available in all institutions, and it is difficult to visualize them with a personal stethoscope.”

Revised: “However, spectrograms may not be available in all institutions, and it is difficult to visualize them with a normal personal stethoscope.”

General comment. 

Comment #7: The research problem must be clearer; authors must support the research idea in the basis of the literature. The scientific background must be stronger.

Comment #8: A hypothesis must be included in the Introduction section.

Response: 

Thank you for your comment.

We were looking for an educational method to improve lung auscultation skills quickly without the need of special devices. Therefore, we hypothesized that simple visualization of lung sounds and writing and visualizing them during auscultation would contribute to a more accurate listening to lung sounds. Our research problem was whether simple graphical and mental visualization of lung sounds improved the auscultation skills of clinical clerkship students, as indicated in the title of the first manuscript. 

(p5. 3rd paragraph of Introduction)

Original: “Herein, we created a simple graphical and mental visualization of lung sounds using a combination of lines and circles in lung auscultation. This study, therefore, aimed to evaluate the effect of educational methods of lung sound auscultation using visualization for medical students in clinical clerkship (CC).”

Revised: “Therefore, we hypothesized that a simplified diagram of lung sounds would improve lung auscultation ability by allowing the learner to visualize lung sounds without special devices.

We created a simple graphical and mental visualization of lung sounds using a combination of lines and circles in lung auscultation. Therefore, this study aimed to evaluate the effect of educational methods of lung sound auscultation using visualization for medical students in clinical clerkship (CC).”

Materials and methods section

Comment #9: Page 4. Line 92. “students attended a one-hour lecture of lung auscultation using a simulator”.

Is it one-hour per week? Or only one-hour for the CC?

Response: 

Thank you for your comment.

As shown in the figure below, this lecture is given only once in the third year before the CC as a preparation for the CC.

Additionally, Mr. Lung® was used for the simulator pointed out in comment #11 in the lecture before CC. Furthermore, Mr. Lung® was also used in the lecture on lung auscultation at CC in Respiratory Medicine.

Accordingly, we revised the related sentence as follows:

(p6, 1st paragraph of Pre-clerkship course and CC in Material and Methods)

Original: “In the third year, before the CC, students attended a one-hour lecture on lung auscultation using a simulator.”

Revised: “In the third year, before the CC, students attended one 60-minute lecture on lung auscultation using a simulator (Mr. Lung®, Kyoto Kagaku Co. Ltd., Kyoto, Japan) as preparation for CC. Mr. Lung® is a mannequin-type lung sounds auscultation simulator widely used in medical schools in Japan (S1A Fig) [9,11].”

Comment #10: Page 5. Line 102. “A total of 89 medical students underwent CC…”

The number of participants must be obtained from the calculation of the sample size “N”, thus authors must include the sample size and how they obtained the sample size.

Response: 

Thank you for your comment.

In this study, the number of participants was decided based on realistic possibilities without calculating the sample size. The sample for this study was recruited based on 120 students in the same academic year who underwent CC at our university hospital in 2020-2021.This is because students in other years only learned about lung auscultation through e-learning due to the COVID-19 pandemic before their CC and had different backgrounds. In addition, the CC in the respiratory unit at our hospital was elective between Respiratory Medicine and Thoracic Surgery. Thus, 89 students rotated through the respiratory medicine department. In addition, students who did not answer all parts of the questionnaire or took the lung auscultation test were excluded from the study. The 18 students who practiced in February and September 2021 during the COVID-19 pandemic in Japan were also excluded from the study since the lectures and tests could not be held due to hospital access restrictions. Therefore, despite the difference in the number of students between the two groups, there was no selection bias in this study.

As you pointed out, the sample size should have been calculated. Furthermore, the actual number of samples evaluated was limited to 65 students for the reasons mentioned above. This has been added to the limitation section.

(p6, 2nd paragraph of Pre-clerkship course and CC in Materials and Methods)

Added: “In this study, the number of participants was based on realistic possibilities without calculating the sample size. Students are rotated from one department to the other every four weeks in the CC of our university hospital. The sample for this study was recruited based on 120 students who underwent the CC in 2020-2021. Students in the following academic years had different backgrounds, as they learned lung auscultation only through e-learning before their CC due to the COVID-19 pandemic. In addition, the CC in the respiratory unit at our hospital was elective between Respiratory Medicine and Thoracic Surgery. Of the 120 students in the same year of the curriculum, 89 chose the Department of Respiratory Medicine.”

(p19, Study limitations)

Original: “First, as it was performed at just one university in Japan, the number of students was small.”

Revised: “First, in terms of research design, this was a single-site study with a small number of participants in an uncontrolled environment and relied partly on students’ self-assessments for data collection. Therefore, a larger number of participants based on the calculation of the required sample size and the use of a reliable questionnaire are required in the future.”

Comment #11: Page 4, line 92. “Students attended a one-hour lecture on lung auscultation using a simulator”.

Is it one-hour per week? Or only one-hour for the CC? What simulation model was used by the students? Give information about the simulation model.

Response: 

As we responded in comment #9, we have revised the related sentence.

Comment #12: Page 4, line 94. Subsequently, each student performed self-learning using a textbook and/or the simulator in preparation for the objective structured clinical examination (OSCE), which is administered in the fall of the fourth year before CC.

Did the authors control the self-learning? Or the authors know the number of hours that each student spent on self-learning with a textbook? Or the authors know the number of hours that each student spent with the simulator in preparation for the OSCE? Because the distinct preparation of each student can influence the results.

Response: 

To prepare for the OSCE, a Japanese textbook and video materials were presented by our university, and the students studied on their own by referring to them. However, this study did not evaluate how the students actually studied for the OSCE. It was difficult to control the students’ self-study and check for their study time through a questionnaire because of the time gap between the preparation for the OSCE and the CC. As stated in response to comment #10, there was no selection bias because the students were divided into the visualization group and the control group according to their rotation period to Respiratory Medicine.

As shown in the S2 Fig, there was no difference between the experience of the two groups in lung auscultation before the CC in Respiratory Medicine and the performance of the CC in Respiratory Medicine based on results other than lung auscultation. 

Therefore, we believe that the influence of the degree of self-study has been neglected in this study.

However, it is an important limitation that we did not investigate the time for self-study for lung auscultation of students before and during the CC. We have added this as a limitation. 

(p19. Study Limitations)

Added: “Second, time spent on lung auscultation education differs between the two groups with and without time spent on training to visualize lung sounds. In addition, the time taken for the self-study of lung auscultation before and during CC can be different for each student.”

Comment #13: Page 5. Line. “A total of 89 medical students underwent CC in Respiratory Medicine at Chiba University Hospital between December 2020 and November 2021”

The number of participants must be obtained from the calculation of the sample size (“N”). Thus, authors must include the sample size and how they obtained the sample size. The reason that the authors had 89 medical students in the CC is not a reason to support the participation of the 89 medical students. The calculation of sample size is mandatory for this type of study.

Response: 

As indicated in our response to comment #10, we based our decision on the number of students in an academic year. As you indicated, it was our fault that we did not calculate the sample size in our study, and we have added a related sentence in response to comment #10.

Comment #14: Page 5. Line 116. “Mr Lung is a mannequin-type lung…”

Authors must include an image showing the Mr. Lung.

Response: 

According to the comment, we added the following picture of Mr. Lung® as S1 Fig (A, B). 

Revised supplementary figure 1.

* The simulator (A, Mr. Lung®, Kyoto Kagaku Co. Ltd., Kyoto, Japan) was used for lectures before and during the CC in Respiratory medicine, the pre- and pos- test (B).

In addition, we have added and revised the related sentence as follows:

(p5, 1st paragraph of Setting in Material and Methods) *Same as change based on comment #9

Original: “In the third year, before the CC, students attended a one-hour lecture on lung auscultation using a simulator.”

Revised: “In the third year, before the CC, students attended one 60-minute lecture on lung auscultation using a simulator (Mr. Lung®, Kyoto Kagaku Co. Ltd., Kyoto, Japan) as preparation for CC. Mr. Lung® is a mannequin-type lung sounds auscultation simulator widely used in medical schools in Japan (S1A Fig) [9,11].” 

(p10. Performance of lung auscultation in Materials and Methods)

Original: “Students’ lung auscultation performance was evaluated by an auscultation test of 10 cases using the Mr. Lung™ simulator.”

Revised: “Students’ lung auscultation performance was evaluated by an auscultation test of 10 cases using the Mr. Lung™ simulator (S1B Fig).”

(p25. Supporting information)

Original: “Supplementary Figure 1. Answer sheet for lung sound auscultation test.

A multiple-choice form (A) was used as the pre-test for the control and visualization group and the post-test for the control group. A multiple-choice form that combined a space in which to draw a figure where there was an abnormality (B) was used as the post-test for the visualization group.”

Revised: “S1 Fig. The simulator (A. Mr. Lung®, Kyoto Kagaku Co. Ltd., Kyoto, Japan) was used for lectures before CC and during CC in Respiratory medicine, the pre- and pos- test (B). Answer sheet for lung sound auscultation test. A multiple-choice form (C) was used as the pre-test for the control and visualization group and the post-test for the control group. A multiple-choice form that combined a space in which to draw a figure where there was an abnormality (D) was used as the post-test for the visualization group.”

Comment #15: Page 5. Line 120. “(Figure 1)”.

The number of participants in each group is a different number (Group Lecture N = 32 vs Group Lecture + Visualizing N = 39). It seems that there is no statistical reason to divide the participants in this no equal division. The problem of this situation comes from the fact that there is not a sample size calculation. The no equal distribution is not helping for getting reliable results.

Response:

As you pointed out, our study is not sufficient in terms of sample size and appropriate grouping due to the number of students in one academic year at our institution and the impact of the COVID-19 pandemic. The fixed number of students in the rotation group in the CC in Respiratory Medicine and restrictions on the CC due to the COVID-19 epidemic resulted in a difference in the number of students between the control group and the visualization group.

Accordingly, the statistical validation of the effect could be a serious limitation of this study. Thus, we have added the following sentence to the limitation section as a Response to Comment #10.

Comment #16: Page 6. Line 126. “The students in the visualization group”

A) In the Introduction section, the authors say that “we created a simple graphical and mental visualization of lung sounds using a combination of lines and circles in lung auscultation”. Authors must explain the procedure for the creation of that graphical tool. It was based on some other graphical images. Was it created by consensus? Did the authors perform any pilot study or experiment to explore if the graphics were easy to understand? The graphics showing the respiratory sounds are the main tool that the authors are using for the teaching-learning, hence, that tool must have a strong scientific background. While this is a nice tool, the authors must detail the origin of this tool.

Response: 

Thank you for your comment.

The diagram showing lung sounds was created based on the spectrogram of lung sounds and the figure presented in the article by Sarker M et al.1 and a Japanese textbook.2 In addition, the diagram was developed by consensus between a Respiratory Medicine specialist (HK) and a medical education specialist (KS).

1. Sarkar M, Madabhavi I, Niranjan N, Dogra M. Auscultation of the respiratory system. Ann Thorac Med 2015;10: 158-68.

2. Institute for Health Care Information Science I. Medical disease : an illustrated reference guide, Respiratory system (3rd Edition): Medic Media, 2018.

The diagrams shown in our study had been used for the education of lung sounds for residents at our university hospital, while it had not been published as a paper.

Accordingly, we have revised the related sentence as follows:

(p. 8-9, 3rd paragraph of Procedure of education on lung sound auscultation with visualization in Material and Methods)

Original: “In the diagram, the vertical axis represents the pitch of lung sounds, and the horizontal axis represents the duration of lung sounds. For respiratory sounds, the thickness of the line represents the loudness of the sound. For the intermittent rales, each crackle is represented by a circle, and the height of the circle’s position indicates the pitch of the sound, while the size of the circle indicates the loudness of the sound. Coarse crackles are presented as circles which are drawn at a lower position on inspiratory and/or expiratory. Fine crackles are presented as circles that are drawn at the end of the inspiratory and are concentrated at a high position. For continuous rales, a horizontal bar was used. The height of the horizontal bar position indicates the pitch of the sound, and the thickness of the bar indicates the loudness of the sound. Wheezes are mainly represented by a horizontal bar at the high end of the expiration, and rhonchi are also mainly represented by a horizontal bar at the low end of the expiration.”

Revised: “The students in the visualization group received approximately 30 minutes of additional training using a diagram of graphically visualized lung sounds during the lecture on lung auscultation in CC, as shown in Fig 2 and S2 Fig. In the diagram, the vertical axis represents the pitch of lung sounds, and the horizontal axis represents the duration of lung sounds as well as a spectrogram of lung sounds. The respiratory sounds were indicated by a right ascending diagonal line for inspiration and a right descending diagonal line for expiration [1]. Moreover, the thickness of the line represents the loudness of the sound. For the intermittent rales, each crackle is represented by a circle, and the height of the circle’s position indicates the pitch of the sound, while the size of the circle indicates the loudness of the sound. The circular shape was chosen as coarse crackles and fine crackles because the former was caused by the bursting of blisters in the bronchi and the latter by the opening of obstructed peripheral bronchi. Thus, coarse crackles were presented as circles which are drawn at a lower position on inspiration and/or expiration. Fine crackles are presented as circles that are drawn at the end of the inspiration and are concentrated at a high position. The height of the horizontal bar position indicates the pitch of the sound, and the thickness of the bar indicates the loudness of the sound as well as the continuous sound shown in the spectrogram. Wheezes are mainly represented by a horizontal bar at the high end of the expiration, and rhonchi are also mainly represented by a horizontal bar at the low end of the expiration. The diagram was created by consensus between Respiratory Medicine specialists (HK) and medical education specialists (KS).”

As you pointed out, it is an important concern whether the diagrams were suitable for students to understand and improve lung auscultation. We will respond to this point in comment #22.

Comment #17: B) Authors must explain in detail the additional training. For instance, did the students watch the graphics in a computer? With a Power Point Presentation? Or in a printed version? How long does the training take? It was a single session? How many times the students drew the representation of the lung sounds? Was it made by hands. What are the indications to mentally visualize lung sounds? Because this is a main section and a main part of the study, the learning with the graphical representation of the lung sounds must be well detailed. Otherwise, the study is not replicable. This teaching method should be tested before the method was applied for a study; the limited description of the method only causes doubts.

Response: 

Thank you for the useful suggestion.

The duration of the lecture on lung auscultation in the CC was one hour, while additional 30 minutes were spent teaching the visualization of lung sound to the visualization groups. The attending doctor taught graphically visualized lung sounds showing diagram slides PowerPoint (Figure 2) on the projector. The diagram figure was the same as the slides, and a blank diagram was distributed to students as PDF files. In addition, during the lecture, the students drew the various lung sounds they listened to over the speakers of the simulator on a blank diagram. After the students drew the lung sounds, the attending physician drew the answer on the whiteboard.

We have also added S2 Fig to provide details on the lecture on lung auscultation in the CC and additional training on the visualization of lung sounds.

(p. 8, 3rd paragraph of Procedure of education on lung sound auscultation with visualization in Material and Methods)

Original: “The students in the visualization group received additional training using a diagram of graphically visualized lung sounds, as shown in Figure 2.”

Revised: “The students in the visualization group received approximately 30 minutes of additional training using a diagram of graphically visualized lung sounds during the lecture on lung auscultation in CC, as shown in Fig 2 and S2 Fig.” 

S2 Fig. Details of the lecture on lung auscultation conducted in the CC and additional training on the visualization of lung sounds.

Unfortunately, the study did not investigate how often students mentally imaged lung sounds during lung auscultation. We have added the following sentences to the paragraphs about increasing the opportunity for drawing lung sounds and repeating instructions to imagine lung sounds to improve the effectiveness of our method.

(p. 18, 5th paragraph in Discussion)

Added: “The study did not investigate how often students mentally imaged lung sounds during lung auscultation. Therefore, it is not clear how many students performed lung auscultation by visualizing lung sounds.”

Comment #18: Page 7. Line 164. “Likert Scale”.

A Likert Scale questionnaire needs of a validation, Cronbach’s alpha must be included to know the reliability of the questionnaire. Hence, this evaluation tool lacks the minimum characteristics of validation.

Response:

Thank you for suggesting the calculation of Cronbach’s alpha.

In this study, we did not calculate Cronbach’s alpha because the questionnaire included one question in each category for the satisfaction with our education method, confidence in lung auscultation, and experience with each lung sound.

As you pointed out, validating the questionnaire is necessary, as shown in Comments #21 and 22. More question items are needed in the questionnaire to analyze what changes occur in medical students regarding lung auscultation through our training. We conducted the new questionnaire with a few students as a preliminary preparation and will calculate Cronbach’s alpha in the future.

Results section

Comment #19: Page 10. Line 209. “Finally, 65 students…”

If authors had a sample size calculation, one could know if this 65 participants is a suitable number for the study.

Response: 

As answered in Comment #10, this study included students who chose the CC in Respiratory Medicine from the 120 students in the one grade at our university. As you mentioned, the sample size should be calculated, and we are planning a new validation with sample size calculations as well.

Comment #20: Page 10. Line 217. “control”

Authors must use same identification name for groups, sometimes they say “lecture group”, other times they say “control group”. Use the same ID in all the manuscript and the reader will not get confused.

Response: 

Thank you for your suggestion. I have unified only the control group.

Comment #21: Page 11. Line 228. “compared to that of the control group”.

Ok, but there were no significant statistical differences. Statistical tests are helpful to accept or reject the hypothesis. When the hypothesis is rejected, is fine, there is no need to give positive interpretation in something that disagree with your hypothesis.

Response: 

Thank you for this comment.

As the reviewer pointed out, there was no significant difference between the two groups with the test results regarding lung auscultation. According to the comment, we have reworded the related sentence as follows:

(p. 12-13, 1st paragraph of Results of tests of lung auscultation in Results)

Original: “Although there was no difference between the scores of the pre- and post-tests of both groups (pre-test, p = 0.827; post-test, p = 0.290), the visualization group scores tended to increase as compared to that of the control group (adjusted mean difference between pre- and post-tests, 3.7±0.5 vs. 3.1±0.6, p = 0.424).”

Revised: “There was no difference between the scores of the pre-and post-tests for both groups (pre-test, p = 0.827; post-test, p = 0.290). In addition, there was no change in scores between the two groups (adjusted mean difference between pre-and post-tests, 3.7±0.5 vs. 3.1±0.6, p = 0.424).”

Discussion section

Comment #21: General comment

The discussion section describes two main findings:

“The visualization of lung sounds can improve medical students” confidence in their lung auscultation.”

“The visualization of lung sounds may improve medical students’ ability to auscultate lung sounds, although the effect is limited.”

The first finding must be explained by the authors in a clear way. This was a finding that support the use of the graphics that represents the lung sound to gain confidence. The authors must explain what learning process occurred in the medical students that gave confidence in their lung auscultation. How can that have happened? It was an effect caused only by the visualization. What mental process might occur? Is there any positive link between the visualization and the training with the simulation model? Does the confidence in their lung auscultation have an impact on the ability to auscultate lung sounds? Some explanation is given in the discussion, but the explanation is more a description of findings from other authors.

Response: 

Thank you for your valuable comment. 

One possible reason for the increase in confidence is that the visualization of lung sounds may have encouraged students to listen more carefully and consciously to the difference between inspiration and expiration and the mechanism behind the lung sounds. Additional clinical information in auscultation can help diagnose and distinguish lung sounds.1,2 Furthermore, it has been suggested that additional visual information, such as a spectrogram, can be useful in lung sound auscultation.3,4 The visualization of lung sounds may have led to lung auscultation being more analytical, considering several elements of lung sounds in inspiration and expiration timing, pitch, and duration. Even if a student could distinguish the lung sound correctly, this multifaceted approach using visualization may have led to the perception of students that the pathophysiology could be better understood and analyzed.

1. Nguyen DQ, Patenaude JV, Gagnon R, Deligne B, Bouthillier I Simulation-based multiple-choice test assessment of clinical competence for large groups of medical students: a comparison of auscultation sound identification either with or without clinical context. Can Med Educ J 2015;6: e4-e13.

2. Shikino K, Ikusaka M, Ohira Y, et al. Influence of predicting the diagnosis from history on the accuracy of physical examination. Adv Med Educ Pract. 2015;6: 143-8.

3. Aviles-Solis JC, Storvoll I, Vanbelle S, Melbye H The use of spectrograms improves the classification of wheezes and crackles in an educational setting. Sci Rep. 2020;10: 8461.

4. Andres E, Gass R Stethoscope: A Still-Relevant Tool and Medical Companion. Am J Med. 2016;129: e37-8.

In addition, the simulation model was used only for lung sound playback during the visualization practice without using a stethoscope in the lecture in the visualization group. Thus, there was no difference between the control and visualization groups in use of the simulator. It is unlikely that the simulator affected the difference in confidence between the two groups in our study. 

However, the detailed process for improving confidence auscultation in students was not completely clear in this study, and further qualitative evaluation, such as focus group interviews and questionnaires, will be considered in future studies.

Accordingly, we have added the related sentences as follows:

(p. 17, 2nd paragraph in Discussion)

Added: “In other words, the visualization of lung sounds may have made lung auscultation more analytical, considering several elements of lung sounds in inspiration and expiration timing, pitch, and duration. Thus, a possible reason for the increase in confidence is that the visualization of lung sounds may have encouraged students to listen more carefully and consciously to the difference between inspiration and expiration and the mechanism behind each lung sounds.”

(p. 17, 3rd paragraph in Discussion)

Added: “Additional clinical information in auscultation can help diagnose and distinguish different lung sounds [27,28]. Furthermore, additional visual information, such as a spectrogram, can be useful in lung sound auscultation [21,22]. Even if students could distinguish the lung sound correctly, this multifaceted approach using visualization may have led to students’ perception that the pathophysiology could be better understood and analyzed. However, the detailed process for improving confidence auscultation in students was not completely clear in this study, and further qualitative evaluation, such as focus group interviews and questionnaires, will be necessary in future studies. Because of the lack of confidence, students may be hesitant to perform lung auscultation on patients.”

Comment #22: The second finding also needs an explanation. It seems that there is no relation between the confidence and the performance on clinics by the medical students. How the authors explain that the visualization of lung sounds had a limited effect on the students’ learning? Maybe the graphics by themselves were the problem. Do the authors know if the graphics were easy to understand? Or if the learning protocol was well-designed to get a positive impact? Is there any relation between the learning with visualization and the learning with the simulation model.

Because the findings are not well explained in the discussion, it seems that the experimental design was not well designed to solve specific question. But more important, it seems that the experimental design was not well designed to explore some hypothesis. Hypothesis is also a guide to understand or to explain the results and evidence. Hence, the discussion must improve to show the scientific value of the study.

Response: 

Thank you for your important comments.

Various factors can have contributed to the lack of improvement in the auscultation skills of the students.

As you pointed out, there can be a problem with the adequacy of the diagram of lung sounds that we developed in our study. The image of lung sounds differs for each person, and it may be necessary to change the diagram to make it more intuitive.1 Therefore, breath sounds may be better shown in a diagram combining triangles rather than lines, as shown in the following figure. Furthermore, it may be easier to intuitively visualize the coarse and fine crackles by using an X instead of a circle as the symbol indicating the crackle in this study as a new S5 Fig. Moreover, we should have evaluated whether the diagram of lung sounds was easy to understand for the students in the questionnaire.

1. Scott G, Presswood EJ, Makubate B, Cross F. Lung sounds: how doctors draw crackles and wheeze. Postgrad Med J. 2013 Dec;89(1058): 693-7.

As answered in comment #21, the range of use of the simulator was almost the same for both groups. The limited use of simulation models makes it difficult to verify the synergistic effects of the visualization of lung sounds. The need to set up a group that only uses visualization without a simulator can be considered.

As you pointed out, the design of our study has been inadequate in various respects, such as the calculation of the sample size, the formulation of the hypotheses, and the preparation of the questionnaire. These issues were affected by the limitation of the students to one academic year and changes in the curriculum due to the COVID-19 pandemic. The CC is being conducted as before the COVID-19 outbreak, and we plan to conduct the research with modifications. We have added these reflections to the discussion of the current study so that they can be linked to the new study.

(p. 18-19, 5th and 6th paragraphs in Discussion) 

* Original texts omitted to save space.

Revised: “This study did not find any significant effect of visualization on lung auscultation. There was no significant difference between the post-test scores of both groups, and there was no change in scores between the two groups. Since post-test scores were only around half the maximum number of points, confidence in lung auscultation will not suffice. There are several possible reasons for the limited effect of our educational method. For the visualization group, we presented a visualization of lung sounds and asked students to draw a diagram of the lung sound that they heard during the lecture. Then, the students were also instructed to imagine lung sounds during their daily practice of lung auscultation. Only instruction to repeatedly visualize lung sounds may be insufficient. The study did not investigate how often students mentally imaged lung sounds during lung auscultation. Therefore, it is not clear how many students performed lung auscultation by visualizing lung sounds. Retention by instructing students to visualize lung sounds during their daily practice may have been more effective. Additional opportunities to draw lung sounds using the simulator or record lung sounds may have enhanced the effectiveness of our educational method. Furthermore, it may be necessary to instruct students to visualize lung sounds repeatedly during daily rounds with their supervisors and provide opportunities to draw visualizations at least once a week. 

Moreover, for effective learning, reflective observation and abstract conceptualization are necessary in addition to active experimentation and concrete experience. Repeating these four steps enhances learning [30]. Malmartel et al. examined the effect of combining a cardiopulmonary auscultation application (Medsounds™, Interactive Systems For Healthcare®) with clinical practice on medical students’ auscultation of heart and lung sounds [17]. Similar to our study, they used a high-fidelity simulator to assess the auscultation of coarse crackles in pneumonia, showing that students’ auscultation performance improved. In addition to the application, they suggested that appropriate supervision from the educator may have had an important impact. 

Another reason for the lack of a significant effect in our study can be that the diagram for the visualization of lung sounds was not appropriate for learners to image mentally. The image of lung sounds can be different for each person; thus, it may be necessary to change the diagram to make it more intuitive [31]. For example, breath sounds may be better shown in a diagram combining triangles rather than lines, as shown in the following figure. Furthermore, it may be easier to intuitively visualize the coarse and fine crackles using an X instead of a circle (S5 Fig). In this study, students in the visualization group drew lung sound diagrams only as an aid during the post-test. In the future, students will be provided with multiple opportunities to draw lung sounds and their lung sound diagrams will be evaluate.”

 

(p25. Supporting information)

Added: “S5 Fig. Proposed changes to lung sound visualization.”

As we mentioned in response to comment #1, we received many very thought-provoking comments this time. These comments improved the content of this manuscript as well as clarified the issues. Therefore, we regard this study as a preliminary study and are now planning to conduct a new study with improvements to address the problems of this study. 

We want to express our gratitude to the reviewers once again.

---

## [Decision Letter · Decision Letter 1]

4 Jan 2023

PONE-D-22-25852R1The effects of simple graphical and mental visualization of lung sounds in teaching lung auscultation during clinical clerkship: A preliminary studyPLOS ONE

Dear Dr. Kasai,

Thank you for submitting your manuscript to PLOS ONE. After careful consideration, we feel that it has merit but does not fully meet PLOS ONE’s publication criteria as it currently stands. Therefore, we invite you to submit a revised version of the manuscript that addresses the points raised during the review process.

We look forward to receiving your revised manuscript.

Kind regards,

Ayse Hilal Bati, Associate Professor

Academic Editor

PLOS ONE

Journal Requirements:

Additional Editor Comments:

Dear Author,

When you review the minor editing suggestions of one of the referees and make the necessary adjustments, it will be possible for me to evaluate your article quickly. I wish you good work.

Reviewers' comments:

Reviewer's Responses to Questions

**Comments to the Author**

1. If the authors have adequately addressed your comments raised in a previous round of review and you feel that this manuscript is now acceptable for publication, you may indicate that here to bypass the “Comments to the Author” section, enter your conflict of interest statement in the “Confidential to Editor” section, and submit your "Accept" recommendation.

Reviewer #1: (No Response)

Reviewer #4: All comments have been addressed

2. Is the manuscript technically sound, and do the data support the conclusions?

Reviewer #1: Yes

Reviewer #4: Yes

3. Has the statistical analysis been performed appropriately and rigorously? 

Reviewer #1: Yes

Reviewer #4: Yes

4. Have the authors made all data underlying the findings in their manuscript fully available?

Reviewer #1: Yes

Reviewer #4: Yes

5. Is the manuscript presented in an intelligible fashion and written in standard English?

Reviewer #1: Yes

Reviewer #4: Yes

6. Review Comments to the Author

Reviewer #1: Thank you for your thoughtful edits to your paper. It is improved, though I still have a few concerns that I'd like you to address in another revision:

Line 65 - change to read "education on how to perform lung.."

Lines 62-82: the order of this paragraph needs to be adjusted. I would suggest moving lines 68-71 to the end of line 64, as you were previously speaking about learning from recordings. This paragraph seems to start about recordings and end with simulators; please restructure it so that it follows that pattern. Also, the sentence about on-the-job training seems inconsistent with the rest of the paragraph which is trying to describe how to actively teach this skill.

Lines 180-183 - state what Q A1 is (can paraphrase) and say it was measured on a scale of 1-5 (define 1 and 5). Then state what A2/A3 measured and state that answers were recorded on a Bohadana et al classification. Readers should only have to refer to the table if they want more info to expand on what's in the text, but what is in the text should be able to be read and understood without the table. In its current form, the text is not understandable without the table.

Lines 180-183 - surveys are stronger when each Likert scale item is defined, but you only defined 1 and 5. This should be added as a limitation to your study.

Table 1 - I would suggest moving this survey to be an appendix rather than a table.

Line 223 - please include the results of the normality testing for each type of data tested (e.g. test scores for lung auscultation; and did you also test the questionnaire answers or did you automatically assume that those were non-parametric because they are ordinal data? Except that you reported mean and SD for the questionnaire answers, which suggests you treated them as parametric data. If you treated ordinal data as non-parametric, then you should have reported median instead of means.)

Line 225 - how and why did you adjust the values?

Lines 239-241 - normality testing results should go in statistical methods not results.

Line 244 - add "between groups" after significant difference.

Lines 296-299 - or they are more confident because they have undergone more training? Students who have completed more training feel more confident because of confirmation bias (if they spend more time on something, their mind wants to confirm that was valuable time spent, so they increase in confidence to perform). You need to include this (more confident due to more instructional time) as a possibility too.

Line 356 - change evaluate to evaluated.

Lines 371-372 - you can't say that it may improve skills (your p value was not significant) but can only say it may improve or does improve confidence.

Reviewer #4: (No Response)

7. PLOS authors have the option to publish the peer review history of their article (what does this mean?). If published, this will include your full peer review and any attached files.

Reviewer #1: No

Reviewer #4: No

---

## [Author Response · Author response to Decision Letter 1]

3 Feb 2023

January 20, 2023

Ayse Hilal Bati, MD, PhD

Academic Editor 

PLOS ONE

Dear Dr. Ayse

Ref. No.: PONE-D-22-25852RI

Thank you for your email dated January 5, 2023, regarding our manuscript, “The effects of simple graphical and mental visualization of lung sounds in teaching lung auscultation during clinical clerkship: a preliminary study.”

We thank you and the reviewers for your insightful comments and insights, based on which we have revised our manuscript. Our point-by-point responses to the reviewers’ comments are attached herewith. The changes to the manuscript are marked in red font.

We believe that our revised manuscript has suitably incorporated the reviewers’ suggestions and is significantly improved over our initial submission. We trust that it is now suitable for publication in PLOS ONE.

Thank you for considering our paper for publication.

Sincerely,

Hajime Kasai, MD

Department of Respirology, Graduate School of Medicine

Chiba University, 1-8-1 Inohana, Chuou-ku Chiba 260-8670, Japan

Telephone: 81-43-222-7171 Ext.71014

Fax: 81-43-226-2176

E-mail: daikasai6075@yahoo.co.jp  

Response to Reviewer #1’s comments

General Comment:

Thank you for your thoughtful edits to your paper. It is improved, though I still have a few concerns that I'd like you to address in another revision:

Response: 

We wish to express our strong appreciation for your insightful comments on our manuscript. The comments have helped us to significantly improve the manuscript. We have marked the relevant changes in red so that you can easily locate them within the manuscript.

Comment #1: Line 65 - change to read "education on how to perform lung.." 

Response: 

According to the comment, we have reworded the related sentence as follows:

(p4. 2nd paragraph of Introduction)

Original: “However, effective education for lung auscultation has not been developed.”

Revised: “However, effective education on how to perform lung auscultation has not been developed.”

Comment #2: Lines 62-82: the order of this paragraph needs to be adjusted. I would suggest moving lines 68-71 to the end of line 64, as you were previously speaking about learning from recordings. This paragraph seems to start about recordings and end with simulators; please restructure it so that it follows that pattern. Also, the sentence about on-the-job training seems inconsistent with the rest of the paragraph which is trying to describe how to actively teach this skill.

Response: 

Thank you for this pertinent comment. 

We agree with you on this point and have adjusted the order of the sentences.

We had placed the sentence about on-the-job training to emphasize the need for more effective learning in medical school, since after graduation, residents will need to learn on their own through on-the-job training. However, as you pointed out, this sentence is inconsistent with the topic of the entire paragraph. Accordingly, we have removed it.

(p4. 2nd paragraph of Introduction)

Removed: “Medical students are often trained in auscultation at school, and after becoming doctors, may find on-the-job training to be their main learning opportunity for lung auscultation.”

Comment #3: Lines 180-183 - state what Q A1 is (can paraphrase) and say it was measured on a scale of 1-5 (define 1 and 5). Then state what A2/A3 measured and state that answers were recorded on a Bohadana et al classification. Readers should only have to refer to the table if they want more info to expand on what's in the text, but what is in the text should be able to be read and understood without the table. In its current form, the text is not understandable without the table.

Response: 

We thank the reviewer for raising this concern.

As you pointed out, it is essential for the contents of the questionnaire to be understood in the body of the text without referring to the table. Accordingly, we have added the following sentences:

(p9, 1st paragraph of Questionnaire in Methods) 

Added: “Students were asked to rate their own lung sound auscultation ability in questions A1 and B1, whether they had any experience in listening for various lung sounds in questions A2 and B2, and their confidence in auscultating lung sounds in questions A3 and B3. In addition, after the CC, they were asked about their satisfaction level with the educational program on lung auscultation in question B4.”

Comment #4: Lines 180-183 - surveys are stronger when each Likert scale item is defined, but you only defined 1 and 5. This should be added as a limitation to your study.

Response: 

Thank you for this comment.

As you pointed out, in the 5-point Likert Scale of the question regarding confidence of lung auscultation in the questionnaires for students, only value 1 and 5 were defined. Thus, the other values were self-judged by the students. More rigorous definitions for all values are needed in future study. 

Accordingly, we have added the related sentence as follows.

(p19-20. Study Limitation)

Original: “The present study has four main limitations. First, in terms of research design, this was a single-site study with a small number of participants in an uncontrolled environment and relied partly on students’ self-assessments for data collection. Therefore, a larger number of participants based on the calculation of the required sample size and the use of a reliable questionnaire are required in the future. Second, time spent on lung auscultation education differs between the two groups with and without time spent on training to visualize lung sounds. In addition, the time taken for the self-study of lung auscultation before and during CC can be different for each student. Third, it is unclear how often students continued to visualize lung sounds in their daily practice. Fourth, in the 5-point Likert Scale of the question regarding to confidence of lung ausculation in the questionnaires for students, only value 1 and 5 were defined. Thus, the other values were self-judged by the students. More rigorous definitions for all values are needed in future study. Fifth, long-term effects have not been explored in our study. ”

Revised: “The present study has five main limitations. First, in terms of research design, this was a single-site study with a small number of participants in an uncontrolled environment and relied partly on students’ self-assessments for data collection. Therefore, a larger number of participants based on the calculation of the required sample size and the use of a reliable questionnaire are required in the future. Second, the time spent on lung auscultation education differs between the two groups with and without time spent on training to visualize lung sounds. This difference in time may have affected students’ confidence in auscultation as a confirmation bias. In addition, the time taken for the self-study of lung auscultation before and during CC can be different for each student. Third, it is unclear how often students continued to visualize lung sounds in their daily practice. Fourth, in the questionnaires for the students, only values 1 and 5 were defined in the 5-point Likert Scale of the question regarding confidence in lung auscultation. The other values were self-judged by the students. Thus, more rigorous definitions for all values are needed in future studies. Fifth, long-term effects have not been explored in our study.” 

Comment #5: Table 1 - I would suggest moving this survey to be an appendix rather than a table.

Response: 

We have changed Table 1 to Supplementary Table 1 and moved it to the Appendix. 

We have also corrected some unforeseen description in questions A3 and B3, as well as a misstatement of the 5-point Likert scale value in responses for question B4 as follows. 

In addition, we have renumbered other tables accordingly.

Revised:

Supplementary table 1. Questionnaire items before and after the training on lung auscultation.

Before the CC in Respiratory Medicine Responses

(A1) How is your current lung sound auscultation ability? Five-point Likert scale

1 (Not confident [cannot listen at all]) to 5 (Confident [able to listen and distinguish between lung sounds and rales])

(A2) Have you ever heard each lung sound from a patient? 

Decreased respiratory sounds, bronchial breathing, prolonged expiration, coarse crackles, fine crackles, wheezes, rhonchi, squawk, and pleural friction rub Yes or No

(A3) How confident are you in auscultating each lung sound?

Same items as in question A2. The response options are same as in question A1.

After the CC in Respiratory Medicine 

(B1) How is your current lung sound auscultation ability? The response options are same as in question A1.

(B2) Have you ever heard each lung sound from a patient?

Same items as in question A2. The response options are same as in question A2.

(B3) How confident are you in auscultating each lung sound?

Same items as in question A2. The response options are same as in question A1.

(B4) What is your level of satisfaction with the education program on lung auscultation? Five-point Likert scale

1 (extremely poor) to 5 (extremely good)

Comment #6: Line 223 - please include the results of the normality testing for each type of data tested (e.g. test scores for lung auscultation; and did you also test the questionnaire answers or did you automatically assume that those were non-parametric because they are ordinal data? Except that you reported mean and SD for the questionnaire answers, which suggests you treated them as parametric data. If you treated ordinal data as non-parametric, then you should have reported median instead of means.)

Response: 

Thank you for this comment.

As shown in the previous response, only scores of lung auscultation tests in control group were normally distributed (Table A). Therefore, we have added Table A excluding rows for age and Results of CC in Respiratory Medicine as Supplementary Table 2.

Table A. P-value in the test of normality of each parameter (n=65).

 Visualization group (n = 35) Control group

(n = 30)

Age ＜0.001 ＜0.001

Results of CC in Respiratory Medicine 0.780 0.012

Satisfaction ＜0.001 ＜0.001

Confidence 

Pre-questionnaire ＜0.001 0.010

Post-questionnaire ＜0.001 ＜0.001

Score of lung auscultation 

Pre-test 0.008 0.610

Post-test 0.049 0.747

We treated all parameters as a nonparametric value to unify the statistical testing methods. Hence, we have presented their medians and interquartile range (IQR) in Table B. 

We have also changed the values in Table 2 (Original: Table 3), as shown in Table B. 

Table B. Satisfaction level with our education and changes in the confidence level pertaining to lung auscultation and the score of lung auscultation before and after education (n = 65). 

 Visualization group (n = 35) Control group

(n = 30) p-value

Satisfaction, median (IQR) 5 (1) 4 (1) 0.150

Confidence in lung auscultation 

Pre-questionnaire, median (IQR) 1 (1) 2 (1) 0.128

Post-questionnaire, median (IQR) 3 (1) 3 (1) 0.028

Mean difference, median (IQR) 2 (1) 1 (1) 0.005

Adjusted mean difference, least mean square (SE) 1.7 (0.1) 1.3 (0.1) 0.020*

Score of lung auscultation 

Pre-test, median (IQR) 11 (2) 11 (5) 0.827

Post-test, median (IQR) 15 (4) 14 (4) 0.290

Mean difference, median (IQR) 4 (4) 3 (1) 0.623

Adjusted mean difference, least mean square (SE) 3.7 (0.5) 3.1 (0.6) 0.424*

* Analysis of covariance (ANCOVA) was performed, and pre-score was adjusted.

Interquartile range, IQR; standard error, SE. 

Accordingly, we have changed the related sentences as follows:

(p2. Result in Abstract)

Original: “Confidence in auscultation of lung sounds significantly increased in both groups (five-point Likert scale, visualization group: pre 1.5±0.1 to post 3.2±0.1, p<0.001; control group: 1.7±0.1 to 2.9±0.1, p<0.001), and was significantly higher in the visualization than in the control group. Moreover, test scores increased in both groups (visualization group: pre-test 11.4±0.5 to post-test 15.1±0.6, p<0.001; control group: 11.1±0.5 to 14.3±0.6, p<0.001).”

Revised: “Confidence in auscultation of lung sounds significantly increased in both groups (five-point Likert scale, visualization group: pre-questionnaire median 1 [Interquartile range 1] to post-questionnaire 3 [1], p<0.001; control group: 2 [1] to 3 [1], p<0.001) and was significantly higher in the visualization than in the control group. Test scores increased in both groups (visualization group: pre-test 11 [2] to post-test 15 [4], p<0.001; control group: 11 [5] to 14 [4], p<0.001).”

(p11. Statistical analysis in Methods section)

Original: “All results are expressed in terms of mean ± standard deviation (SD), unless otherwise indicated.”

Revised: “All results are expressed in terms of the median and interquartile range (IQR) unless otherwise indicated.”

(p12. 1st paragraph of Result section)

Original: “The visualization group included 35 students (age 23.2±1.6 years; male/female 27/8), and the control group included 30 students (22.6±0.8 years; male/female 24/6) (Fig 1). There were no significant differences in the age and sex ratio (p=0.123 and p=0.780, respectively). In addition, there was no significant difference in the results of CC in Respiratory Medicine, which did not include the results of the lung auscultation test between the two groups (maximum score: 25, 76.7±3.9 vs. 77.1±5.1, p=0.802).”

Revised: “The visualization group included 35 students (age median 23 [IQR 2]; male/female 27/8), and the control group included 30 students (22 [IQR 1] years; male/female 24/6) (Fig 1). There were no significant differences in the age and sex ratio (p=0.123 and p=0.780, respectively). In addition, there was no significant difference in the results of CC in Respiratory Medicine, which did not include the results of the lung auscultation test between the two groups (maximum score: 100, 76.4 [5.8] vs. 76.5 [5.4], p=0.802).”

(p12. 1st paragraph of Questionnaire in Result section)

Original: “Based on an analysis of their responses, students’ satisfaction level with our education was high in both groups, but there was no significant difference (visualization group: 4.5±0.7 vs. control group: 4.2±0.9, p = 0.150).”

Revised: “Based on an analysis of their responses, students’ satisfaction level with our education was high in both groups, but there was no significant difference between groups (visualization group: median 5 [IQR 1] vs. control group: 4 [1], p = 0.150).”

(p13. 2nd paragraph of Questionnaire in Result section)

Original: “Confidence in lung auscultation (A1 and B1: “How is your current lung sound auscultation ability?”) significantly increased in both groups (five-point Likert scale; visualization group, pre-test: 1.5±0.6 to post-test: 3.3±0.7, p < 0.001; control group, 1.7±0.6 to 2.9±0.8, p < 0.001). Confidence in lung auscultation after CC was found to be significantly higher in the visualization group than in the control group (p = 0.028; adjusted mean difference between pre- and post-test, 1.7±0.1 vs. 1.3±0.1, p = 0.020).”

Revised: “Confidence in lung auscultation (A1 and B1: “How is your current lung sound auscultation ability?”) significantly increased in both groups (five-point Likert scale; visualization group, pre-questionnaire: 1 [1] to post-questionnaire: 3 [1], p < 0.001; control group, 2 [1] to 3 [1], p < 0.001). Confidence in lung auscultation after CC was found to be significantly higher in the visualization group than in the control group (p = 0.028; adjusted mean difference between pre- and post-questionnaire, 1.7 [standard error 0.1] vs. 1.3 [0.1], p = 0.020).”

(p13. 1st paragraph of Results of tests of lung auscultation in Result section)

Original: “The test score increased in both groups (visualization group, pre- 11.4±2.5 to post-test 15.1±3.5, p < 0.001; control group, 11.1±3.3 to 14.3±3.2, p < 0.001) as shown in Table 3. There was no difference between the scores of the pre-and post-tests for both groups (pre-test, p = 0.827; post-test, p = 0.290). In addition, there was no change in scores between the two groups (adjusted mean difference between pre-and post-tests, 3.7±0.5 vs. 3.1±0.6, p = 0.424).”

Revised: “The test score increased in both groups (visualization group, pre-test 11 [2] to post-test 15 [4], p < 0.001; control group, 11 [5] to 14 [4], p < 0.001) as shown in Table 2. There was no difference between the scores of the pre-and post-tests for both groups (pre-test, p = 0.827; post-test, p = 0.290). In addition, there was no change in scores between the two groups (adjusted mean difference between pre-and post-tests, 3.7 [standard error 0.5] vs. 3.1 [0.6], p = 0.424).”

Comment #7: Line 225 - how and why did you adjust the values?

Response: 

Thank you for this query.

In this study, confidence in and scores on lung auscultation in the two groups before and after the lecture were compared using analysis of covariance (ANCOVA), by adjusting the values of the pre-questionnaire and pre-test results for each group. The change in the values of the pre-questionnaire and pre-test results was adjusted as an outcome for the comparison. Moreover, these adjustments were performed to avoid any misalignment of the values of each parameter when comparing the groups.

Accordingly, we have revised the related sentences as follows:

(p11-12, Statistical analysis in Methods)

Original: “Confidence in and scores on lung auscultation in the two groups before and after our lecture were compared using analysis of covariance (ANCOVA), by adjusting the values of the pre-questionnaire and pre-test results for each group.”

Revised: “Confidence in and scores on lung auscultation in the two groups before and after our lecture were compared using analysis of covariance (ANCOVA), by adjusting the values of the pre-questionnaire and pre-test results for each group. Change of confidence in and scores on lung auscultation in the two groups before and after our lecture were compared using multivariate linear regression. In the multivariate analysis, the values of the pre-questionnaire and pre-test results were adjusted.”

Comment #8: Lines 239-241 - normality testing results should go in statistical methods not results.

Response: 

Thank you for this important suggestion. 

Accordingly, we have moved the sentences as follows:

(p11. Statistical analysis in Methods)

Original: “All results are expressed in terms of mean ± standard deviation (SD), unless otherwise indicated. The Wilcoxon signed-rank test was used to compare the questionnaire answers about participants’ confidence in lung auscultation and the test scores of lung auscultation before and after the training. Normality was evaluated using the Shapiro-Wilk test. Confidence in and scores on lung auscultation in the two groups before and after our lecture were compared using analysis of covariance (ANCOVA), by adjusting the values of the pre-questionnaire and pre-test results for each group. A p-value < 0.05 was considered statistically significant. All statistical analyses were performed using JMP 15.0 software (Cary, North Carolina, USA) and SAS software version 9.4 (SAS Institute, Cary, USA).”

Revised: “All results are expressed in terms of the median and interquartile range (IQR) unless otherwise indicated. Normality was evaluated by performing the Shapiro–Wilk test. In the data collected in this study, the pre- and post-tests for the control group were normally distributed, while the other parameters were non-normally distributed (Supplementary table 2). The Wilcoxon signed-rank test was used to compare the questionnaire answers about participants’ confidence in lung auscultation and the test scores of lung auscultation before and after the training. Confidence in and scores on lung auscultation in the two groups before and after our lecture were compared using analysis of covariance (ANCOVA), by adjusting the values of the pre-questionnaire and pre-test results for each group. Change of confidence in and scores on lung auscultation in the two groups before and after our lecture were compared using multivariate linear regression. In the multivariate analysis, the values of the pre-questionnaire and pre-test results were adjusted. A p-value < 0.05 was considered statistically significant. All statistical analyses were performed using JMP 15.0 software (Cary, North Carolina, USA) and SAS software version 9.4 (SAS Institute, Cary, USA).”

Comment #9: Line 244 - add "between groups" after significant difference.

Response: 

According to your suggestion, we have reworded the relevant sentence as follows:

(p12. 1st paragraph of Questionnaire in Result section)

Original: “Based on an analysis of their responses, students’ satisfaction level with our education was high in both groups, but there was no significant difference (visualization group: 4.5±0.7 vs. control group: 4.2±0.9, p = 0.150).”

Revised: “Based on an analysis of their responses, students’ satisfaction level with our education was high in both groups, but there was no significant difference between groups (visualization group: median 5 [IQR 1] vs. control group: 4 [1], p = 0.150).”

Comment #10: Lines 296-299 - or they are more confident because they have undergone more training? Students who have completed more training feel more confident because of confirmation bias (if they spend more time on something, their mind wants to confirm that was valuable time spent, so they increase in confidence to perform). You need to include this (more confident due to more instructional time) as a possibility too.

Response: 

Thank you for this comment.

As you noted, the time for lecture on lung auscultation was longer in the visualization group. The additional time added to the common part of the lecture on lung sound auscultation for the two groups was 30 minutes, which was a relatively short time for the duration of the 4-week CC. Moreover, we did not inform the participants that the lecture time was different between the visualization and control groups. 

However, the possibility of the influence of confirmation bias on the improvement of confidence cannot be ruled out. 

Accordingly, we have added it as the third limitation under Study Limitation.

(p19, Study limitation)

Original: “Second, time spent on lung auscultation education differs between the two groups with and without time spent on training to visualize lung sounds. In addition, the time taken for the self-study of lung auscultation before and during CC can be different for each student.”

Revised: “Second, the time spent on lung auscultation education differs between the two groups with and without time spent on training to visualize lung sounds. This difference in time may have affected students’ confidence in auscultation as a confirmation bias. In addition, the time taken for the self-study of lung auscultation before and during CC can be different for each student. ”

Comment #11: Line 356 - change evaluate to evaluated.

Response: 

According to the comment, we have changed the related sentence as follows:

(p18. 7th paragraph of Discussion)

Original: “In this study, students in the visualization group drew lung sound diagrams only as an aid during the post-test. In the future, students will be provided with multiple opportunities to draw lung sounds and their lung sound diagrams will be evaluate.”

Revised: “In this study, students in the visualization group drew lung sound diagrams only as an aid during the post-test. In the future, students will be provided with multiple opportunities to draw lung sounds and their lung sound diagrams will be evaluated.”

Comment #12: Lines 371-372 - you can't say that it may improve skills (your p value was not significant) but can only say it may improve or does improve confidence.

Response: 

Thank you for this suggestion.

As you pointed out, our educational program could only improve students’ confidence in lung auscultation. Accordingly, we have revised the related sentence as follows:

(p19. Conclusion)

Original: “The visualization of lung sounds may improve medical students’ lung auscultation skills and their confidence in auscultation.”

Revised: “The visualization of lung sounds may improve medical students’ confidence in lung auscultation.”

---

## [Editor Report · Decision Letter 2]

14 Feb 2023

The effects of simple graphical and mental visualization of lung sounds in teaching lung auscultation during clinical clerkship: A preliminary study

PONE-D-22-25852R2

Dear Dr. Hajime Kasai,

We’re pleased to inform you that your manuscript has been judged scientifically suitable for publication and will be formally accepted for publication once it meets all outstanding technical requirements.

Kind regards,

Ayse Hilal Bati, Associate Professor

Academic Editor

PLOS ONE

Additional Editor Comments (optional):

Dear Authors,

Thank you for making the suggested changes to your article accordingly.
---

## [Editor Report · Acceptance letter]

8 Mar 2023

PONE-D-22-25852R2 

The effects of simple graphical and mental visualization of lung sounds in teaching lung auscultation during clinical clerkship: A preliminary study 

Dear Dr. Kasai:

I'm pleased to inform you that your manuscript has been deemed suitable for publication in PLOS ONE. Congratulations! Your manuscript is now with our production department. 

Kind regards, 

on behalf of

Dr. Ayse Hilal Bati 

Academic Editor

PLOS ONE